# Communicating doctors' consensus persistently increases COVID-19 vaccinations

Vojtěch Bartoš[1,2 ✉], Michal Bauer[3,4], Jana Cahlíková[5] & Julie Chytilová[3,4]

The reluctance of people to get vaccinated represents a fundamental challenge to containing the spread of deadly infectious diseases[1,2], including COVID-19. Identifying misperceptions that can fuel vaccine hesitancy and creating effective communication strategies to overcome them are a global public health priority[3–5]. Medical doctors are a trusted source of advice about vaccinations[6], but media reports may create an inaccurate impression that vaccine controversy is prevalent among doctors, even when a broad consensus exists[7,8]. Here we show that public misperceptions about the views of doctors on the COVID-19 vaccines are widespread, and correcting them increases vaccine uptake. We implement a survey among 9,650 doctors in the Czech Republic and find that 90% of doctors trust the vaccines. Next, we show that 90% of respondents in a nationally representative sample (n = 2,101) underestimate doctors' trust; the most common belief is that only 50% of doctors trust the vaccines. Finally, we integrate randomized provision of information about the true views held by doctors into a longitudinal data collection that regularly monitors vaccination status over 9 months. The treatment recalibrates beliefs and leads to a persistent increase in vaccine uptake. The approach demonstrated in this paper shows how the engagement of professional medical associations, with their unparalleled capacity to elicit individual views of doctors on a large scale, can help to create a cheap, scalable intervention that has lasting positive impacts on health behaviour.

COVID-19 is a salient example of a disease with profound economic, social and health impacts, which can be controlled by large-scale vaccination if enough people choose to be vaccinated. Nevertheless, a large percentage of people are hesitant to get a vaccine, preventing many countries from reaching the threshold necessary to achieve herd immunity[9,10]. Consequently, rigorous evidence on scalable approaches that can help to overcome people's hesitancy to take a COVID-19 vaccine is a global policy priority[3–5]. Existing research has made important progress in documenting the roles of providing financial incentives[11,12], reminders[4,5], information about the efficacy of the vaccines[13,14], the role of misinformation[15] on the intentions of the public to get vaccinated and, more recently, also on their actual decisions to get a vaccine[5] shortly after an intervention. However, little is known about whether cheap, scalable strategies with the potential to cause lasting increases in people's vaccination demand and uptake exist. A focus on the persistence of the impacts of interventions is especially important for vaccines such as those against COVID-19, which are often distributed in phases to different demographic groups due to capacity constraints, and multiple doses spaced over time are required to avoid declines in protection.

In many surveys across the globe, people report that they strongly trust the views of doctors[6]. This makes it crucial to understand how people perceive doctors' views about the COVID-19 vaccine. In this paper, we pursue the hypothesis that reluctance to adopt the vaccine originates, in part, in misperceptions about the distribution of aggregate views of the medical community: many people may fail to recognize that there is a broad consensus in favour of the vaccine among doctors. Furthermore, we argue and show that professional associations can serve as aggregators of individual views in a medical community, by helping to implement surveys eliciting the views of doctors on a large scale. Disseminating information of a broad consensus, when one exists, can lead to people updating their perceptions of doctors' views and, in turn, may induce lasting changes in vaccination demand and uptake.

Our focus on public misperceptions of the views of doctors is motivated by a widespread concern that media coverage can create uncertainty and polarization in how people perceive expert views, even when a broad consensus actually exists. In terms of traditional media, a desire to appear neutral often motivates journalists to provide a 'balanced' view by giving roughly equal time to both sides of an argument[7,16], creating an impression of controversy and uncertainty[8]. Such 'falsely balanced' reporting has been shown to be a characteristic element of policy debates ranging from climate change[7,16] to health issues, including links between tobacco and cancer, and potential side effects of vaccines[8,17]. In the context of the COVID-19 vaccines, casual observation suggests that media outlets often feature expert opinions that highlight the efficacy of approved COVID-19 vaccines together with skeptical experts who voice concerns about rapid

[1]Department of Economics, University of Munich, Munich, Germany. [2]Department of Economics, Management and Quantitative Methods, University of Milan, Milan, Italy. [3]CERGE-EI, a joint workplace of Charles University and the Economics Institute of the Czech Academy of Sciences, Prague, Czech Republic. [4]Institute of Economic Studies, Faculty of Social Sciences, Charles University, Prague, Czech Republic. [5]Department of Public Economics, Max Planck Institute for Tax Law and Public Finance, Munich, Germany. ✉e-mail: vojtech.bartos@unimi.it

vaccine development and untested side effects. The media usually do not specify which claims are supported by the wider medical community, leading the World Health Organization to warn media outlets against engaging in false-balance reporting[18]. Furthermore, polarization of beliefs can arise due to echo chambers—people choosing to be exposed to expert opinions or opinion programmes that fuel their fears of the vaccine or, alternatively, to those who strongly approve of it[19–21].

We study these issues in the Czech Republic, which is a suitable setting, given the observed level of vaccine hesitancy among a large share of its population, similar to the situation in many other countries. At the time of data collection, the acceptance rate of the vaccine in the Czech Republic was around 65%, compared to 55–90% in other countries globally. At the same time, the Czech Republic ranks close to the median level of trust and satisfaction with medical doctors, based on a comparison of 29 countries[6]. We provide more background in Section 3.1 of the Supplementary Information.

We start by documenting and quantifying public misperceptions about the views of doctors on the COVID-19 vaccines. Shortly before the COVID-19 vaccine rollout began, we implemented a short online survey among 9,650 doctors. We found strong evidence of consensus: 90% of doctors intend to get vaccinated themselves and 89% trust the approved vaccines. At the same time, we found evidence of systemic and widespread misperceptions of the views held by the medical community among a nationally representative sample of the adult population ($n$ = 2,101): more than 90% of people underestimate doctors' trust in the vaccines and their vaccination intentions, with most people believing that only 50% of doctors trust the vaccines and intend to be vaccinated.

These findings set the stage for our main experiment, in which we tested whether randomized provision of information about the actual views of doctors can recalibrate public beliefs and, more importantly, cause a lasting increase in vaccination uptake. The experimental design aimed to make progress on two important empirical challenges that are common in experiments on the determinants of demand for COVID-19 vaccines. First, as an intention–behaviour gap has been documented in the context of flu vaccines and other health behaviours[22], measuring both vaccination intentions and actual vaccination uptake allows us to test whether treatment effects on vaccination intentions translate into behavioural changes of a similar magnitude. The initial set of studies on COVID-19 vaccination, typically implemented before the vaccines became available, only tested impacts on intentions[11,14,15], although recent exceptions exist[5,23].

Second, most experiments designed to correct misperceptions about the views of others, and other information provision experiments in various domains, including migration, health and political behaviour, document treatment effects to be substantially smaller when measured with a delay[24,25]. In theory, the worry is that individual perceptions about the views of doctors might shift between the time when the treatment takes place and when people decide whether to actually get vaccinated, for reasons including regression of perceptions to the mean, biased recall or motivated memory[26]. Conversely, researchers have suggested that providing facts about a widely shared consensus of trustworthy experts might be resilient to these forces[17], as the treatment may reduce incentives to seek new information, and condenses complex information into a simple fact ('90% of doctors trust the approved vaccines'), which is easy to remember. Understanding whether providing information about medical consensus has temporary or lasting effects on vaccination demand is informative for policy, in terms of whether a one-off information campaign is sufficient, or whether the timing of messages needs to be tailored for different groups of people who become eligible for a vaccine at different points in time, and also whether such an information campaign needs to be repeated in cases of multiple-dose vaccines.

To address these issues, our experiment is integrated into longitudinal data collection with low attrition rates. The treatment was implemented in March 2021. We used data from 12 consecutive survey waves collected from March to November 2021, covering the early period when the vaccine was scarce, later when it gradually became available to more demographic groups, and finally for several months when it was easily available to all adults. This is reflected in the vaccination rates, which increased in our sample from 9% in March to 20% in May and to nearly 70% in July. Then, it grew slowly to 77% at the end of November. This longitudinal, data-collection-intensive approach allows us to estimate: (1) whether disseminating information on the consensus view of the medical community has immediate effects on people's beliefs and their intentions to get the vaccination shortly after the intervention; (2) whether the effects translate into actually getting vaccinated, even though most of the participants became eligible for the vaccine only many weeks after the intervention; and (3) whether the effects on vaccine uptake are persistent or whether the vaccination rate of untreated individuals eventually catches up, perhaps due to ongoing governmental campaigns, stricter restrictions for individuals who are not vaccinated, or greater potential life disruptions during severe epidemiological periods.

## Consensus of the medical community

We conducted a supplementary survey to gather the views of doctors on COVID-19 vaccines in February 2021. The survey was implemented in partnership with the Czech Medical Chamber (CMC), whose contact list includes the whole population of doctors in the country, because membership is compulsory. All doctors who communicate with the CMC electronically (70%) were asked to participate and 9,650 (24% of those contacted) answered the survey. Supplementary Table 1 provides summary statistics and documents that the sample is quite similar, in terms of age, gender, seniority and location, to the overall population of medical doctors in the Czech Republic.

Figure 1 shows the distribution of doctors' responses. A clear picture arises, suggesting that a broad consensus on COVID-19 vaccines exists in the medical community: 89% trust the vaccine (9% do not know and 2% do not trust it), 90% intend to get vaccinated (6% do not know and 4% do not plan to get vaccinated) and 95% plan to recommend that their patients take a vaccine (5% do not). These responses are broadly similar across gender, age, years of medical practice and size of the locality in which the doctors live: for all sub-groups, we found the share of positive answers to all questions ranges between 85% and 100% (Supplementary Table 2). Using probability weights based on observable characteristics of the entire population of doctors in the country makes very little difference in the estimated distribution of opinions in our survey. Reassuringly, the opinions in our survey are in line with high actual vaccination rates (88%) observed among Czech doctors when vaccines became available[27], despite vaccination not being compulsory for any profession, including for doctors.

## Longitudinal experiment

Our main sample consists of participants in the longitudinal online data collection 'Life during the pandemic', organized by the authors in cooperation with PAQ Research; the data were collected by the NMS survey agency (Methods and Supplementary Methods). The information intervention was implemented on 15 March 2021 (wave 0). We used data from 12 consecutive waves of data collection regularly conducted from March to November 2021. This time span covers the period when the vaccination was gradually rolled out and eligibility rules changed regularly, making the vaccine available for more demographic groups (until June 2021), and a period when vaccination was freely available for the entire adult population (from July 2021).

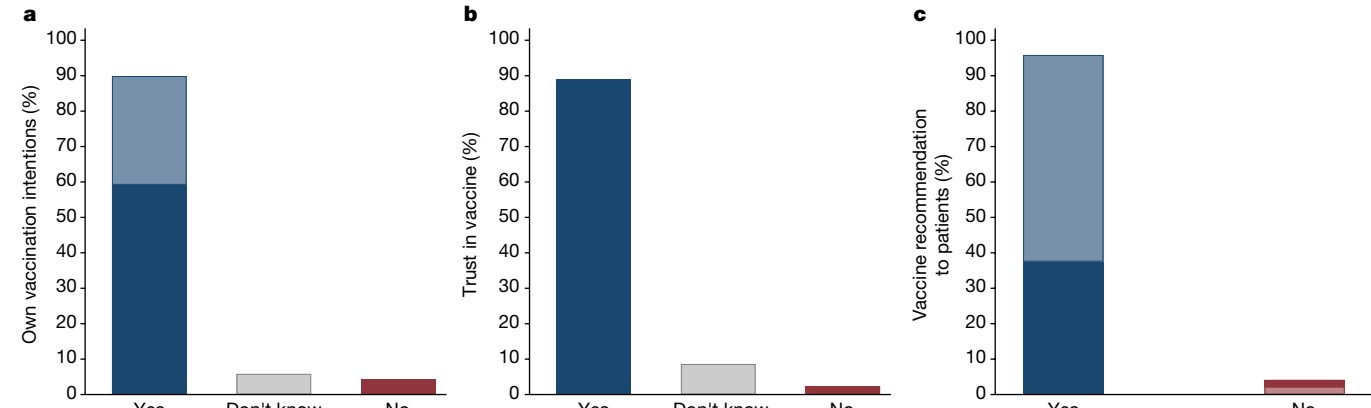

**Fig. 1 | The views of doctors on COVID-19 vaccines.** Supplementary study among the members of the CMC ($n = 9,650$). **a**, Distribution of responses to the question "Will you personally be interested in getting vaccinated, voluntarily and free of charge, with an approved vaccine against COVID-19?". Among participants who answered yes, the dark blue refers to those who reported already being vaccinated, whereas the light blue refers to those who plan to get vaccinated. **b**, Responses to the question "Do you trust COVID-19 vaccines that have been approved by the European Medicines Agency (EMA) approval process?". **c**, Responses to the question "Will you recommend COVID-19 vaccination to your healthy patients to whom you would recommend other commonly used vaccines?" Among participants who answered yes, the dark blue refers to those who would recommend the vaccines even without being asked, whereas the light blue refers to those who would recommend only when asked. In Supplementary Table 2, we show that the distribution of views is similar across various demographic groups and level of seniority.

The sample from wave 0 is our 'base sample' ($n = 2,101$). By design, the sample is broadly representative of the adult Czech population in terms of a host of observable characteristics (for summary statistics, see Extended Data Table 1). In addition, the vaccination rate reported in our sample closely mimics the levels and dynamics of the overall adult vaccination rate in the country (Extended Data Fig. 1). This comparison suggests that attitudes to vaccination in our sample are likely to be representative of the larger population, in contrast to surveys based on convenience samples[28]. Although this pattern is reassuring, we cannot test and fully rule out a possibility that our sample might not be representative in terms of unobservable characteristics affecting receptivity to the information treatment studied. Furthermore, the response rate in the follow-up waves is high, ranging between 76% and 92%. A large portion of participants ($n = 1,212$; the 'fixed sample') took part in all 12 waves of data collection.

The participants were randomly allocated to either the Consensus condition ($n = 1,050$) or Control condition ($n = 1,051$) in wave 0. In the Consensus condition, they were provided with a summary of the survey among medical doctors, including three charts that displayed the distribution of doctors' responses regarding their trust in the vaccines, willingness to get vaccinated themselves and intentions to recommend the vaccine to patients. In the Control condition, the participants did not receive any information about the survey of medical doctors and only filled the regular part of the longitudinal survey.

In all 12 waves, we asked whether respondents got vaccinated against COVID-19. The main outcome variable 'vaccinated' is equal to one if the respondent reported having obtained at least one dose of a vaccine against COVID-19. We also elicited prior beliefs on the views of doctors about the vaccines in wave 0 shortly before the information intervention, and posterior beliefs in wave 1 2 weeks afterwards.

Extended Data Table 1 and Supplementary Table 3 show no systematic differences in the set of baseline characteristics pre-registered as control variables. Nevertheless, because the randomization was not stratified on baseline covariates, there are random imbalances in some covariates, as expected. Some of the larger differences are for variables not included in the set of pre-registered control variables. Specifically, before the intervention, compared to participants in the Control condition, the individuals in the Consensus condition were slightly less likely to be vaccinated themselves (standardized mean difference (SMD) = 0.069), and expected a smaller percentage of doctors to trust the vaccine (SMD = 0.072) or to intend to get vaccinated (SMD = 0.090). As these three variables are highly predictive of vaccination uptake, we report two main regression specifications: (1) with the pre-registered set of control variables, and (2) with control variables selected by the LASSO procedure[29]. To document robustness, we also report estimates with no control variables and with alternative sets of control variables.

## Misperceptions about doctors' views

To quantify misperceptions about the views of doctors on COVID-19 vaccines, we compared the prior beliefs of participants about doctors' views, measured before the intervention, with the actual views of the doctors from the CMC survey. We found strong evidence of misperceptions. The average, median and modal guesses are that 57%, 60% and 50% of doctors, respectively, want to be vaccinated (Fig. 2a), whereas in reality 90% of doctors do. The average, median and modal guesses about the percentage of doctors who trust the vaccines are 61%, 62% and 50%, respectively (Fig. 2b), whereas in practice 89% of doctors report trusting the vaccines. A vast majority of participants underestimate the percentage of doctors who want to be vaccinated (90%) and those who trust the vaccines (88%).

The distribution of beliefs reveals that the large underestimation does not originate in two distinct groups of participants holding opposite views of the medical consensus—one group thinking that most doctors have positive views about the vaccines and the other group thinking that most doctors are skeptical about them. Instead, most people expect a wide diversity of attitudes across individual doctors. Of participants, 81% believe that the percentage of doctors who want to be vaccinated is between 20% and 80%. For beliefs about doctors' trust in the vaccines, this number is 76%. Furthermore, these misperceptions are widespread across all demographic groups based on age, gender, education, income and geographical regions (Supplementary Table 4).

We found several intuitive descriptive patterns that increase confidence in our measures of beliefs. First, beliefs about the vaccination intentions of doctors and their trust in the vaccines are strongly positively correlated ($r(2,099) = 0.60$, $P < 0.001$). Second, beliefs about doctor's trust and vaccination intentions are highly predictive of respondents' own intentions and uptake (Supplementary Table 4). In the next sub-section, we explore whether this relationship is causal. Third, in Supplementary Fig. 1, we show that misperceptions about the

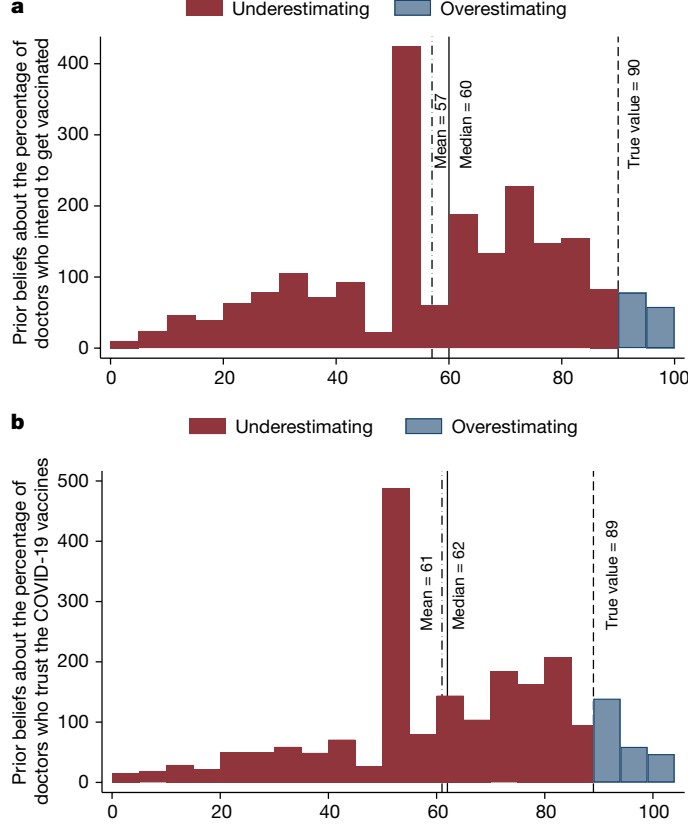

**a**

Legend: ■ Underestimating ■ Overestimating

y-axis: Prior beliefs about the percentage of doctors who intend to get vaccinated

Mean = 57
Median = 60
True value = 90

**b**

Legend: ■ Underestimating ■ Overestimating

y-axis: Prior beliefs about the percentage of doctors who trust the COVID-19 vaccines

Mean = 61
Median = 62
True value = 89

**Fig. 2 | Perceptions of doctors' views on COVID-19 vaccines.** A sample of the adult Czech population (*n* = 2,101). **a**, Distribution of the prior beliefs of respondents about what percentage of doctors would like to get vaccinated. **b**, Distribution of the beliefs of respondents about what percentage of doctors trust approved COVID-19 vaccines. The dashed line shows the true value, based on the responses of doctors in the Supplementary study. The red and blue colours show the percentage of those who underestimate and overestimate, respectively, doctors' own vaccination intentions (**a**) and trust in the COVID-19 vaccines (**b**).

doctor's views are unlikely to arise due to the inattention of participants to the questions. The results are very similar when we excluded the 4% of participants who did not pass all of the attention checks embedded in the survey, and when we excluded the 10% of participants with the shortest response times.

## Intervention impacts on vaccination

We first established the effects of the intervention on posterior beliefs about the views and vaccination intentions of doctors shortly after the intervention. We found that the information provided shifts expectations about the views of doctors (Fig. 3a and Supplementary Table 5). Two weeks after the intervention (in wave 1), the Consensus condition increased beliefs about the share of doctors who trust the vaccines by 5 percentage points (p.p.) (*P* < 0.001) and beliefs about the share of doctors who want to get vaccinated by 6 p.p. (*P* < 0.001). Next, the Consensus condition increased the prevalence of people intending to get vaccinated by around 3 p.p. (*P* = 0.039; Fig. 3b and Supplementary Table 6). When we restricted the sample to those who participated in all waves, we found the point estimate to be slightly larger (5 p.p., *P* = 0.001).

Next, we found a systematic, robust and lasting treatment effect on vaccine uptake. Four months after the intervention, when vaccines became available to all adults, we found that participants in the Consensus condition were around 4 p.p. more likely to be vaccinated than those in the Control condition (Figs. 4 and 5). As expected, owing to

the gradual rollout of the vaccine during the March to June period, the effect emerged gradually (Extended Data Table 2 provides more information about changes in vaccine eligibility rules). The difference in the uptake rates between the Consensus and Control conditions steadily increased to 4–5 p.p. in July and remained relatively stable thereafter (Fig. 4 and Extended Data Table 3).

In Fig. 5 and Extended Data Table 4, we report results from pooled regressions to utilize data from all six waves implemented in July to November, include wave fixed effects and cluster standard errors at the individual level. The estimated treatment effect is significant for both main specifications—when we control for a set of variables selected by the LASSO procedure (*P* = 0.005) and when we control for the pre-registered set of variables (*P* = 0.026). The effect is similar when estimated in each of these waves separately (Fig. 4).

The estimated effect size is slightly larger (4.4 p.p.) when we used the specification with LASSO-selected control variables than when we used the specification with pre-registered control variables (3.5 p.p.). Figure 5 shows that this is because the LASSO procedure selects baseline beliefs and vaccination status as relevant control variables, whereas these variables are not included in the pre-registered set. Consequently, both approaches document robust positive treatment effect between 3.5 and 4.4 p.p. Readers who believe that researchers should control for random imbalances in important baseline variables may favour the upper bound, whereas readers concerned about departures from pre-registered analyses may favour the lower bound.

Our finding of a positive treatment effect does not rely on a specific choice of control variables or estimation strategy. First, the effect is very similar when we controlled for various sets of baseline variables other than the pre-registered and LASSO-selected sets, as well as when we controlled for none (Fig. 5 and Extended Data Table 4). Second, the effect is significant at conventional levels when we calculated *P* values using the randomization inference method (Extended Data Tables 3 and 5). Third, the estimated treatment effect is 5.4 p.p. (*P* = 0.008) when we used baseline data about vaccination rates, and used a difference-in-difference estimation (Supplementary Table 7). Furthermore, the results are robust to excluding participants who arguably paid less attention (Extended Data Table 5). As in the analysis of vaccination intentions, the estimated effects on uptake are slightly larger when we restricted the analysis to those who participated in all 12 waves.

Differential attrition cannot explain our findings. First, we found that the participation rate is relatively high and does not differ across the Consensus and Control conditions on average. There is also no evidence of differential attrition by baseline covariates, suggesting that different types of individuals were not participating in the Consensus and Control conditions (Supplementary Table 8). We found this pattern for participation in each of the 11 follow-up waves separately as well as when we focused on participation in all waves (being in the fixed sample). As a sensitivity test, we imputed missing vaccination status for those who did not participate in some of the waves and assumed either that (1) their vaccination status has not changed since the last wave for which the data are available, or that (2) their status is the same as the one reported in the earliest next wave for which the data are available. The first approach allowed us to impute all the missing information because we know the vaccination status of each participant in the initial wave. The second approach allowed us to impute the missing information, except in cases when a respondent did not participate in the last wave. The effects are robust (Extended Data Table 5).

The effect of the Consensus condition on uptake is lasting. First, although in the main estimates we focused on the likelihood of respondents getting at least one vaccine dose, a qualitatively similar and significant effect emerges when we focused on the likelihood of participants getting two doses (Extended Data Fig. 2). Second, the treatment effect emerges during a 3-month period, due to availability restrictions, and then is stable across all six follow-up waves covering the July to November period (Fig. 4). Thus, the main effect is not driven by differences in

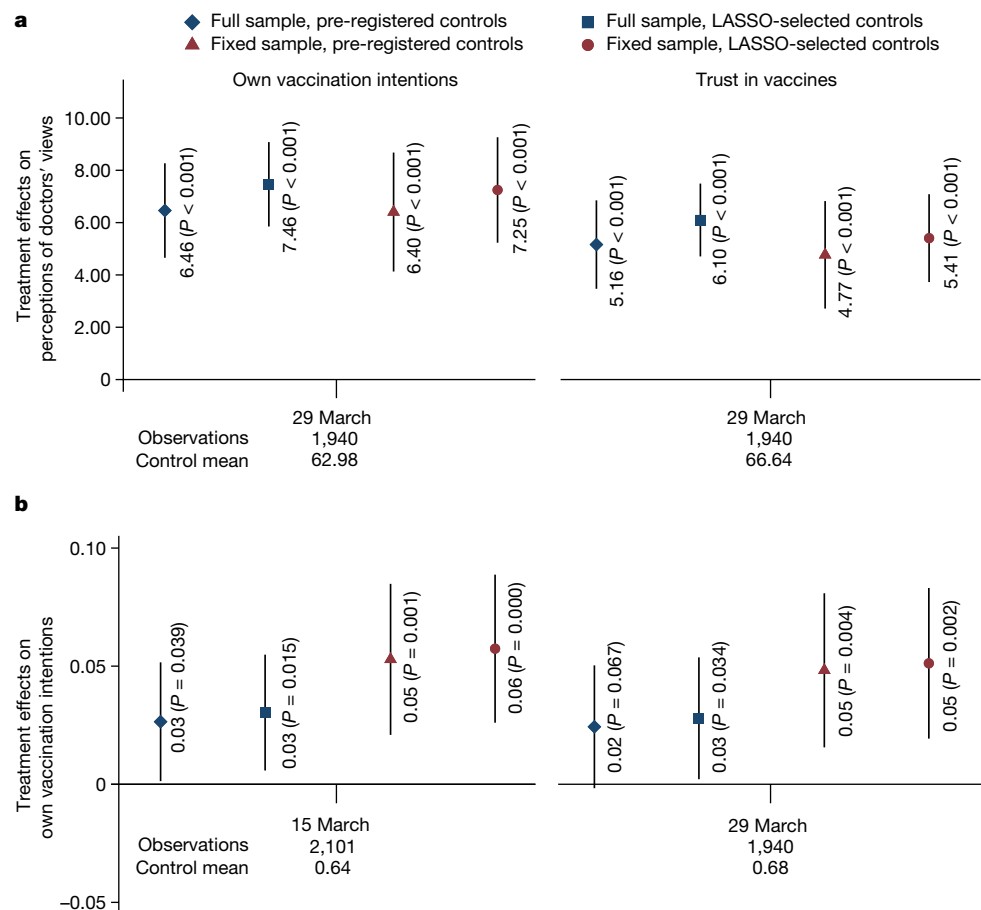

**Fig. 3 | Effects of the Consensus condition on posterior beliefs about doctors' views and vaccination intentions.** A sample of the adult Czech population. **a**, Estimated effects of the Consensus condition on beliefs about the percentage of medical doctors who plan to get vaccinated (left panel) and on beliefs about the percentage of doctors who trust approved COVID-19 vaccines (right panel), measured in wave 1 (29 March; Consensus condition $n = 970$; Control $n = 970$). **b**, The dependent variable is an indicator for an intention to be vaccinated with a vaccine against COVID-19, measured in wave 0 (15 March; Consensus condition $n = 1,050$; Control $n = 1,051$) and wave 1 (29 March; Consensus condition $n = 970$; Control $n = 970$). We report the results of two specifications: (1) a linear probability regression controlling for pre-registered covariates: gender, age category (6 categories), household size, number of children, region (14 regions), town size (7 categories), education (4 categories), economic status (7 categories), household income

(11 categories) and baseline vaccination intentions, and (2) a double-selection LASSO linear regression selecting from a wider set of controls in Extended Data Table 1, including prior vaccine uptake and beliefs about the views of doctors. Markers show the estimated effects and the whiskers denote the 95% confidence interval based on Huber–White robust standard errors. The estimated effects and Student's $t$-test (two-sided) $P$ values are reported in the figure. No adjustments were made for multiple comparisons. We report estimates for (1) all observations, full sample (diamond and square), and (2) for a sub-sample of participants who took part in all 12 waves (Consensus condition $n = 614$; Control $n = 598$), fixed sample (triangle and circle). In the lower part of the figure, we report the timing, the total number of observations and the Control mean for each wave. See Supplementary Section 3.5 for further specification details. Supplementary Tables 5 and 6 show the regression results for **a** and **b** in detail, respectively.

the timing of getting vaccinated. Last, in the September and November waves, we asked about the intentions of participants to get a booster dose. The estimated effect is very similar in magnitude as the effect on uptake of the first dose (around 4 p.p.), suggesting that the information intervention elevates vaccination demand even 9 months after it was implemented (Extended Data Fig. 2).

Documenting such persistence has interesting implications. As the demand for vaccination in the Control condition does not catch up with the Consensus condition over such a long period, the results suggest that the type of vaccine hesitancy reduced by the Consensus condition is resilient to policies, campaigns or any life disruptions that participants were exposed to during the period studied. This includes a severe COVID-19 wave that took place in November 2021 in the Czech Republic, which resulted in one of the highest national mortality rates in global comparisons (see Section 3.1 of the Supplementary Information and Extended Data Fig. 3).

The point estimates of around 4 p.p. imply a relatively large effect size, especially in light of the low costs of the intervention. As the

vaccination rate in the Control condition was 70–75% during the July to November period, the Consensus condition reduces the number of those who are not vaccinated by 13–16%. To compare, providing truthful information about the vaccination intentions of other people was shown to increase intentions to get vaccinated by 1.9 p.p.[30]. Nudging health workers to get vaccinated by referring to vaccinated colleagues has been shown to increase the likelihood of their registering for vaccination by around 3 p.p.[31]. More generally, the most successful, low-cost behavioural nudges with documented effect on uptake have estimated effect sizes up to 5 p.p.[4,5], which is quite similar to the effect of providing information about consensus in doctors' opinions studied here. In addition, a noteworthy aspect of our study is the documented persistence of the effects, which is another crucial margin for assessing the intervention effectiveness.

The Supplementary Information describes exploratory analyses of how the treatment effect differs across different sub-samples of respondents (Supplementary Table 5 and Extended Data Table 5). Reassuringly, we found that the positive effect on vaccine uptake is

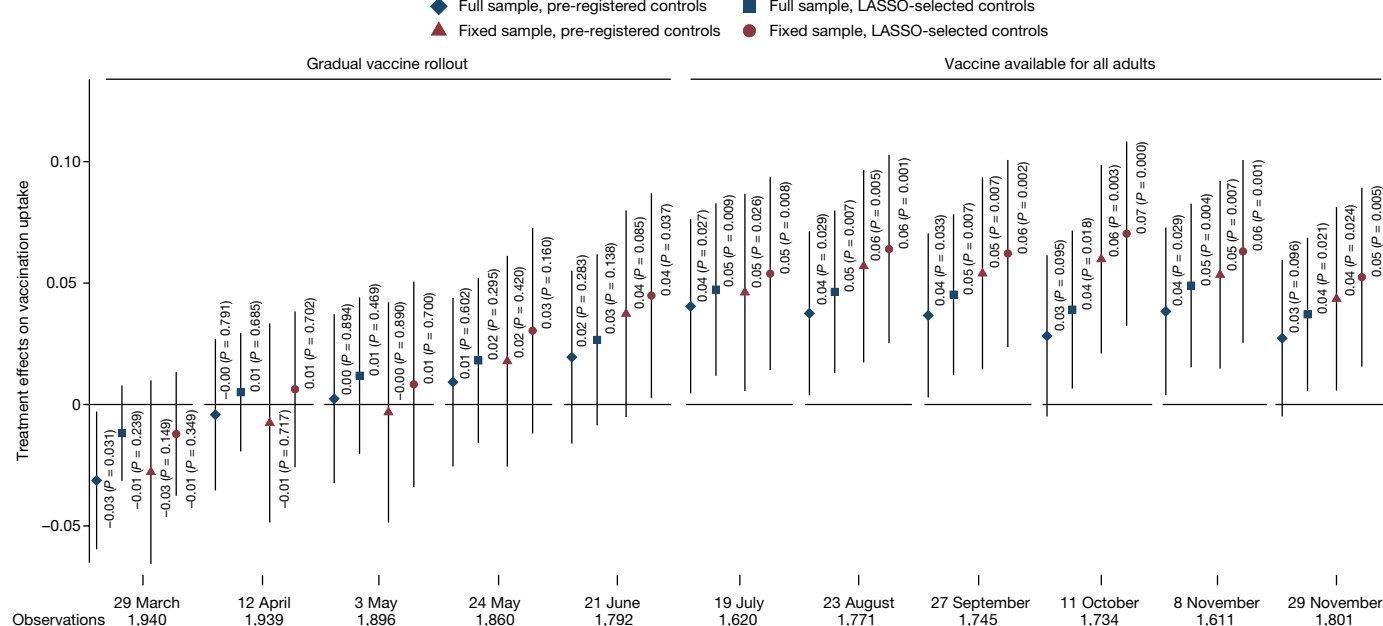

**Fig. 4 | Effects of the Consensus condition on vaccination uptake.** A sample of the adult Czech population. Estimated effects of the Consensus condition by survey wave on getting at least one dose of a vaccine against COVID-19. We report the same four specifications as in Fig. 3 (linear probability model with pre-registered controls using full (diamond) and fixed (triangle) samples, and double-selection LASSO linear regression selecting from controls in Extended Data Table 1 using full (square) and fixed (circle) samples). Markers show the estimated effects and the whiskers denote the 95% confidence interval based on Huber–White robust standard errors. The estimated effects

and Student's *t*-test (two-sided) *P* values are reported in the figure. No adjustments were made for multiple comparisons. We report estimates for (1) all observations, full sample (diamond and square), and (2) for a sub-sample of participants who took part in all 12 waves, fixed sample (triangle and circle). In the lower part of the figure, we report the timing, the total number of observations and the Control mean for each wave. Full sample: Consensus condition *n* = 807–970, Control *n* = 800–973; see Extended Data Table 2 for exact *n* per wave. Fixed sample: Consensus condition *n* = 614; Control *n* = 598. Extended Data Table 3 shows the regression results in detail.

concentrated among those who underestimated doctors' trust and vaccination intentions, whereas no systematic effect was observed among overestimators. In addition, the effect is driven by those who initially did not intend to get vaccinated, in line with the interpretation that the intervention changed the views of individuals who were initially skeptical about the vaccine. Nevertheless, the analysis of heterogenous effects should be treated as tentative because the differences in coefficients are not always significant and we did not adjust for testing of multiple hypotheses.

Given that vaccination status is self-reported, we provide several tests documenting that the observed effect does not arise due to priming or the experimenter demand motivating some people in the Consensus condition to report being vaccinated even when they were not. We begin by noting that the observed treatment effect is lasting and emerged only gradually over several months, as more people became eligible to get vaccinated. By contrast, priming and experimenter demand effects are typically thought to be relevant mainly for responses shortly after a treatment[25,32].

To probe more directly, we used two distinct approaches to verify the reported vaccination status in the main dataset. First, inspired by existing work[25,33], we used additional data about vaccination status collected for us by a third, independent party among the same sample. As the survey agency, graphical interface and topic of the survey were different from our main data collection, the experimenter demand effect that might be potentially associated with treatment in our main survey is unlikely to affect responses in the third-party verification survey. Only two respondents (one in the Consensus condition and one in the Control condition) reported being vaccinated in the main survey, but reported the opposite in the verification survey (Extended Data Table 6), so mismatch in reporting of being vaccinated is very rare in general and not related to treatment. We arrive at a similar conclusion

using the second verification approach that links reported vaccination status with an official proof of vaccination: an EU Digital COVID certificate issued by the Czech Ministry of Health. We showed that respondents in the Consensus condition compared to the Control condition are not less willing or able to provide verifiable information from the certificate (Extended Data Table 6). Finally, we showed that the effect of the Consensus condition on lower prevalence of those reporting not being vaccinated in the main survey is almost fully explained by greater prevalence of those reporting being vaccinated and having their vaccination status verified (Supplementary Table 9). More details about the methods and results of both verifications appear in the Methods section and in Section 3.4 of the Supplementary Information.

## Discussion

Our results shed light on the role that misperceptions of the distribution of expert views have in vaccine hesitancy, and also show how this barrier can be lifted by providing accurate information. We provide evidence that (1) the vast majority of medical doctors in the Czech Republic trust the approved COVID-19 vaccines, (2) the vast majority of respondents in a nationally representative survey substantially underestimate the percentage of doctors with positive views of the vaccine, and (3) correcting these misperceptions has lasting positive effects on vaccine uptake. Although existing experiments have made progress in identifying low-cost strategies to increase vaccination intentions[4,13–15] and uptake[5] measured shortly after the intervention, this paper integrates the experiment in longitudinal online data collection and contributes by identifying a low-cost, scalable treatment that has lasting effects on behaviour.

Scientists, and the medical community as a whole, have invested enormous efforts to develop and deliver COVID-19 vaccines. However,

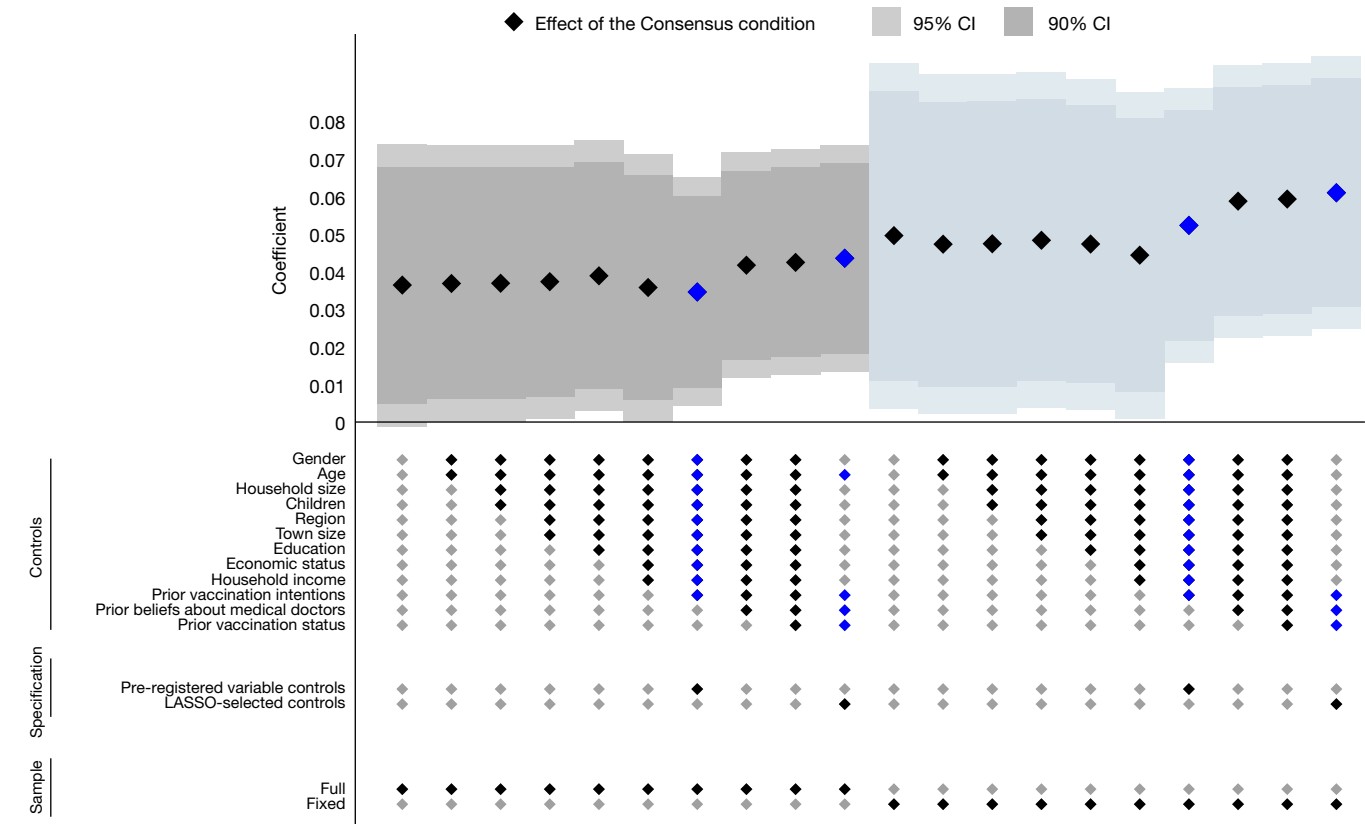

**Fig. 5 | Effects of the Consensus condition on vaccine uptake: robustness.**
A sample of the adult Czech population. This specification chart plots the estimated effects of Consensus on the likelihood of vaccine uptake for a pooled sample across waves 6–11 (when the vaccine was available for all adults). All specifications include wave fixed effects. Markers show the estimated effects, the darker or lighter whiskers denote the 90% or 95% confidence interval, respectively, based on standard errors clustered at the respondent level. No adjustments were made for multiple comparisons. We report a range of linear probability model specifications by sequentially adding sets of control variables in Extended Data Table 1. The main specifications are marked by blue diamonds. We report all specifications for both the full sample (left-hand side) and the fixed sample (right-hand side). Full sample: Consensus condition $n = 5,145$ (981 clusters = respondents); Control $n = 5,137$ (983 clusters = respondents). Fixed sample: Consensus $n = 3,684$ (614 clusters = respondents); Control $n = 3,588$ (598 clusters = respondents). Extended Data Table 4 shows the regression results in detail.

much less collective effort has been directed at informing the public of the high levels of trust in the vaccine across the broad medical community. Here we show that professional medical associations can serve as aggregators of individual doctors' views, by facilitating opinion polls among doctors. Resulting data can be used in campaigns to tackle vaccine hesitancy and also as input for media reports. Although we cannot empirically pin down the sources of the misperceptions observed in our study, we suspect that they originate, at least in part, in a journalistic norm in which balance is often considered a mark of objective and impartial reporting, and a way to attract the attention of news consumers[34]. Our results strengthen the case for supplementing contrasting views on controversial issues with information about how prevalent such views are[35].

To guide efforts to scale up this intervention, we discuss what types of factors may affect its efficiency and how we view the boundary conditions in terms of the applicability of the intervention beyond the context that we studied. We estimate the effects of a one-time intervention, among a sample in which most people probably paid attention to the information. Understanding whether the efficiency of the intervention can be fostered by repeated provision of information, as some research has suggested[36], and which modes of delivery, such as media advertisements, text messages or informational mail flyers, can best attract a sufficient degree of attention is an important next step for future research. Next, in many settings, implementing such information campaigns by governments, health insurance companies or healthcare providers may help to facilitate access to the contacts of large numbers

of individuals[4,5] and to address the need for a trusted source to provide the information intervention. Furthermore, in theory, this type of intervention should have larger effects: (1) the greater the trust in medical doctors in a given country is, and (2) the greater the prevalence of misperceptions about the views of doctors towards a vaccine is. We studied this intervention in a country with an approximately median level of trust in doctors[6], which provides some confidence that our findings from the Czech Republic may extend to other settings. At the same time, to our knowledge, because this is the first paper to provide direct evidence of the prevalence and size of misperceptions about the views of doctors on COVID-19 vaccines, we can only speculate how widespread such misperceptions are in other settings. Given that the likely sources of the misperceptions—false-balance reporting and echo chambers—are not specific to the Czech Republic, and given that misperceptions about scientific consensus have been documented in other countries in other domains, including health and climate change[24,37], we suspect that this bias in beliefs about COVID-19 vaccines is relatively widespread. We hope to see more research on this front.

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

# Article

## Methods

### Supplementary survey among doctors

To gather the views of doctors on COVID-19 vaccines, we implemented a survey in partnership with the CMC, to maximize coverage of the medical community. The survey was implemented online in February 2021. Because membership in the CMC is compulsory, the CMC has a list of contacts for the whole population of doctors in the country. The CMC approached all doctors who communicate with the CMC electronically (70%) and asked them to participate in a short survey, using the Qualtrics platform. Of doctors contacted, 9,650 (24%) answered the survey. The doctors in our sample work in all regions of the country, are on average 52 years of age, 64% are female individuals and 62% have more than 20 years of experience. A comparison of characteristics of doctors in our sample and of all doctors in the Czech Republic is presented in Supplementary Table 1.

### Main experiment

**Sample.** Our main sample consisted of 2,101 participants of the longitudinal online data collection 'Life during the pandemic', organized by the authors in cooperation with PAQ Research and the NMS survey agency. In March 2020, the panel began to provide real-time data on developments in economic, health and social conditions during the COVID-19 pandemic. We used data from 12 consecutive waves of data collection conducted at 3–4-week intervals between mid-March and the end of November 2021.

The information intervention was implemented on 15 March 2021, which we labelled as wave 0. The sample from wave 0 is the 'base sample' ($n = 2,101$, 1,052 female participants and 1,049 male participants, mean age of 52.9 years (s.d. = 15.98), youngest 18 years of age, oldest 92 years of age). The base sample is broadly representative of the adult Czech population in terms of sex, age, education, region, municipality size, employment status before the COVID-19 pandemic, age × sex, and age × education. Prague and municipalities with more than 50,000 inhabitants are oversampled (boost 200%). Sample statistics are presented in Extended Data Table 1. The sample is close to being representative of the adult Czech population in terms of attitudes to COVID-19 vaccines. The development of the proportion of people getting vaccinated in the Control condition very closely mimics the actual vaccination rates in the Czech Republic (Extended Data Fig. 1), when we weighted the observations in our sample to be representative in terms of observable characteristics.

An important feature of the panel is that participants agreed to be interviewed regularly, and the response rate is high throughout the study: it ranges between 76% and 92% in individual follow-up waves, and is 86% for the last wave, implemented at the end of November 2021. Of participants, 1,212 (58%) took part in all 12 waves of data collection: they form the 'fixed sample'. Consequently, in the analysis, we report the main results for (1) all participants from the base sample who responded in a given wave, which we denote 'full sample', and for (2) the 'fixed sample', composed of individuals who participated in all 12 waves, eliminating the potential role of differences in samples across waves and making it easier to gauge the dynamics of treatment effects.

**Information intervention.** In wave 0, the participants were randomly assigned to either the Consensus condition ($n = 1,050$) or the Control condition ($n = 1,051$). In the Consensus condition, they were informed that the CMC conducted a large survey of almost 10,000 doctors from all parts of the country to collect their views on COVID-19 vaccines. They were also informed that the views were similar for doctors of different genders, ages and regions. Then, the participants were shown three charts displaying the distribution of responses of doctors regarding their trust in the vaccines, willingness to get vaccinated themselves and intentions to recommend the vaccine to their patients. Each of the charts was supplemented by a short written summary. The exact wording and the charts are provided in Section 3.3 of the Supplementary Information. In the Control condition, the participants did not receive any information about the survey of medical doctors.

**Data.** Before the information intervention in wave 0, we elicited prior beliefs about doctor's views to quantify misperceptions about doctors' opinions. Specifically, the participants were asked to estimate (1) the percentage of doctors in the Czech Republic who trust the approved vaccines, and (2) the percentage of doctors who are either vaccinated or intend to get vaccinated themselves. Later, in wave 1, we elicited posterior beliefs to estimate whether people in the Consensus condition actually updated their beliefs about doctors' views based on the information provided. In each of the 12 waves, we asked respondents to report whether they got vaccinated against COVID-19. The main outcome variable 'vaccinated' is equal to one if the respondent reported having obtained at least one dose of a vaccine against COVID-19.

In the analysis, we report two main regression specifications: (1) a linear probability regression controlling for pre-registered covariates: gender, age (6 categories), household size, number of children, region (14 regions), town size (7 categories), education (4 categories), economic status (7 categories), household income (11 categories) and prior vaccination intentions, and (2) a double-selection LASSO linear regression selecting from a wider set of controls in Extended Data Table 1, including prior vaccine uptake and beliefs about the views of doctors.

### Additional data to verify vaccination status

We collected two sets of additional data to verify the reported vaccination status in the main dataset.

**Third-party verification.** First, we used data collected for us by a third, independent party. We took advantage of the fact that different survey agencies have access to the panel our respondents are sampled from (the Czech National Panel). Although the main data collection was implemented by one agency (NMS), we partnered with another agency (STEM/MARK) to include a question on vaccination status in a survey implemented on its behalf among the same sample. As the survey agency, graphical interface and topic of the survey were different from our main data collection, we believe that respondents considered the two surveys to be completely independent of each other, and thus experimenter demand unlikely had a role in the second survey. The response rate was high (92.8%) and independent of the treatment (Extended Data Table 6). Out of 1,801 participants in wave 11, 1,672 also took part in the third-party verification survey implemented 2 weeks later. This allowed us to compare reported vaccination status at the individual level for a vast majority of our sample, and to test whether Consensus affects the level of consistency in reporting of being vaccinated across surveys.

**Certificate verification.** The second verification links the reported vaccination status with an official proof of vaccination. We exploited the fact that all vaccinated people receive an EU Digital COVID certificate issued by the Czech Ministry of Health, which was often used as a screening tool at the time of data collection. We collected the data on vaccination certificates among respondents from our full sample who (1) participated in wave 11, and (2) reported to have at least one dose of the COVID-19 vaccine in wave 11 ($n = 1,414$). We asked respondents whether they had the certificate with them. Of participants, 96% confirmed that they had the certificate with them, and this proportion is very similar across the Consensus and Control conditions ($\chi^2(1, n = 1,414) = 0.999, P = 0.318$). Those with a certificate were asked to type in several specific pieces of information about the applied vaccine that are unlikely to be known by someone without a certificate (for example, the correct answer for those who got a vaccine from Pfizer/Biontech is 'SARS-CoV-2 mRNA'). Assessment of the typed text by independent raters suggests that, conditional on their having the certificate, more than 94% of respondents actually looked at the certificate when responding to our detailed questions. This rate is again very similar across conditions ($\chi^2(1, n = 1,364) = 0.473, P = 0.492$).

More details about both verification procedures and results are in the Supplementary Information.

## Ethics approval
The research study was approved by the Commission for Ethics in Research of the Faculty of Social Sciences of Charles University. Participation was voluntary and all respondents provided their consent to participate in the survey.

## Reporting summary
Further information on research design is available in the Nature Research Reporting Summary linked to this paper.

## Data availability
The experiment and analyses were pre-registered on the AEA RCT Registry (AEARCTR-0007396). The dataset generated and analysed for the main experiment is available in the Harvard Dataverse repository (https://doi.org/10.7910/DVN/RH0T6R). The availability of the dataset from the supplementary survey with medical doctors is subject to the approval of the CMC.

## Code availability
The code to replicate the analyses and figures is available in the Harvard Dataverse repository (https://doi.org/10.7910/DVN/RH0T6R).

38. Komenda, M. et al. Complex reporting of coronavirus disease (COVID-19) epidemic in the Czech Republic: use of interactive web-based application in practice. *J. Med. Internet Res.* **22**, e19367 (2020).

**Acknowledgements** We thank PAQ Research (especially D. Prokop) and NMS Market Research (especially L. Rambousek) for implementing the longitudinal data collection; STEM/MARK for implementing the third-party verification; the CMC, M. Kubek, D. Valášek, M. Matoušek, R. Mounajjed and J. Studený for implementing the supplementary survey of medical doctors and providing further statistics; and D. Cantoni, F. Matějka, G. Veramendi and J. Zápal for feedback. V.B. thanks the German Research Foundation (CRC TRR 190 and 444754857). M.B. and J.Chytilová thank the Czech Science Foundation (20-11091S). J.Cahlíková thanks the Max Planck Institute for Tax Law and Public Finance for generous funding of the project. This project has received funding from the European Research Council (grant agreement no. 101002898). The funders had no role in the design of the study, data collection and analysis, decision to publish or preparation of the manuscript. D. Korlyakova, T. Bielaková, M. Pospíšil, I. Burianová and D. Nováková provided excellent assistance.

**Author contributions** V.B., M.B., J.Cahlíková and J.Chytilová contributed equally to the preparation of the design, implementation and interpretation of the findings from the main experiment. V.B. initiated the project, organized and implemented the survey among doctors, and had a lead role in the data analyses. M.B. and J.Chytilová wrote the manuscript. J.Cahlíková secured most of the funding.

**Competing interests** The authors declare no competing interests.

**Additional information**
**Correspondence and requests for materials** should be addressed to Vojtěch Bartoš.

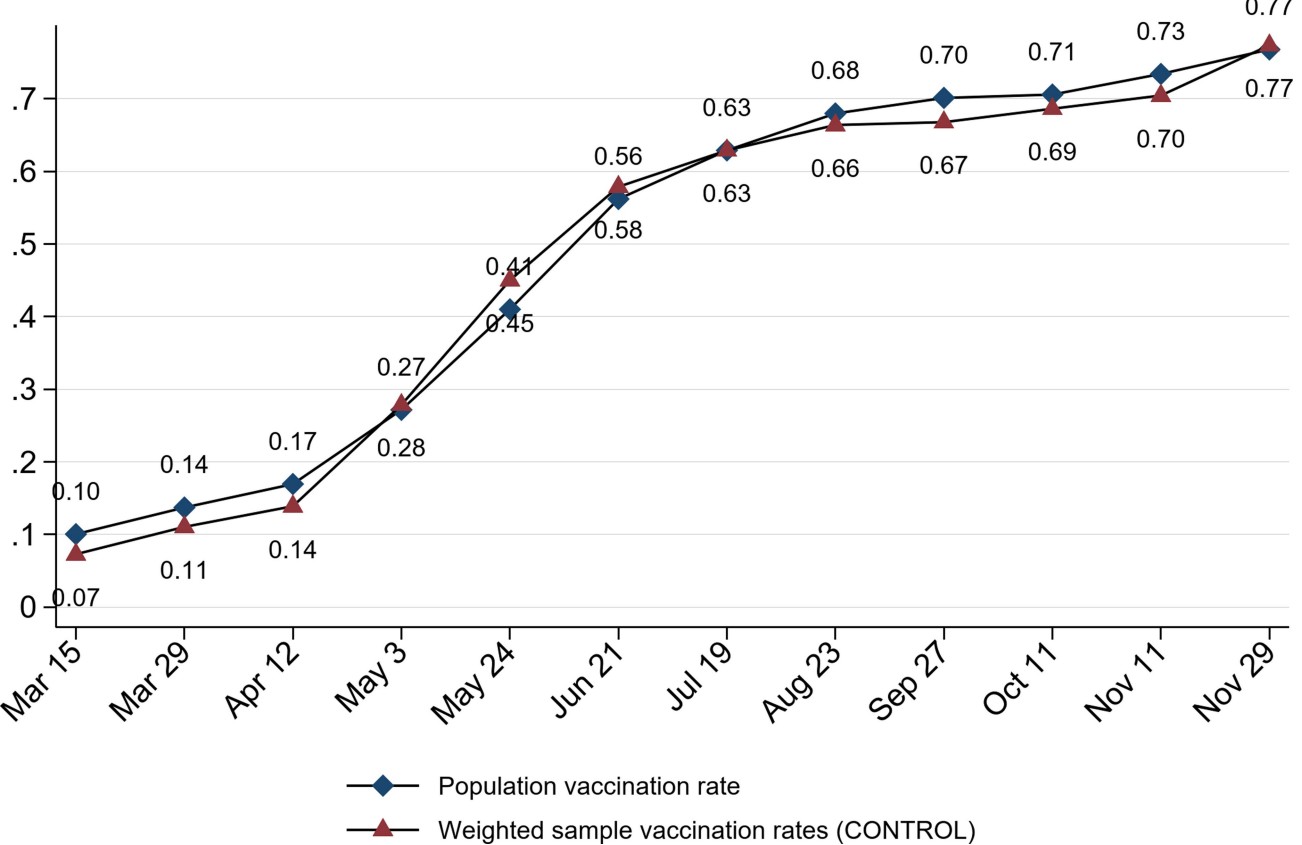

**Extended Data Fig. 1 | Comparison of development of vaccination rate in the Control group (Sample of adult Czech population) and the Czech adult population.** The horizontal axis represents a timeline. Population data means are for a Tuesday following the start of the data collection (Mondays) at a respective wave denoted by diamonds. The weighted Control group means are denoted by triangles. Control condition n = 800–1,051, depending on survey wave. Source of population data: Opendatalab, a website set up by the Faculty of Information Technologies at the Czech Technical University in Prague using open data from the Czech Ministry of Health (https://ockovani.opendatalab. cz/statistiky), ISSN 2787-9925 · http://aleph.techlib.cz/F/?func=direct&doc_number=000017426&local_base=STK02 (accessed on January 12, 2022)[38].

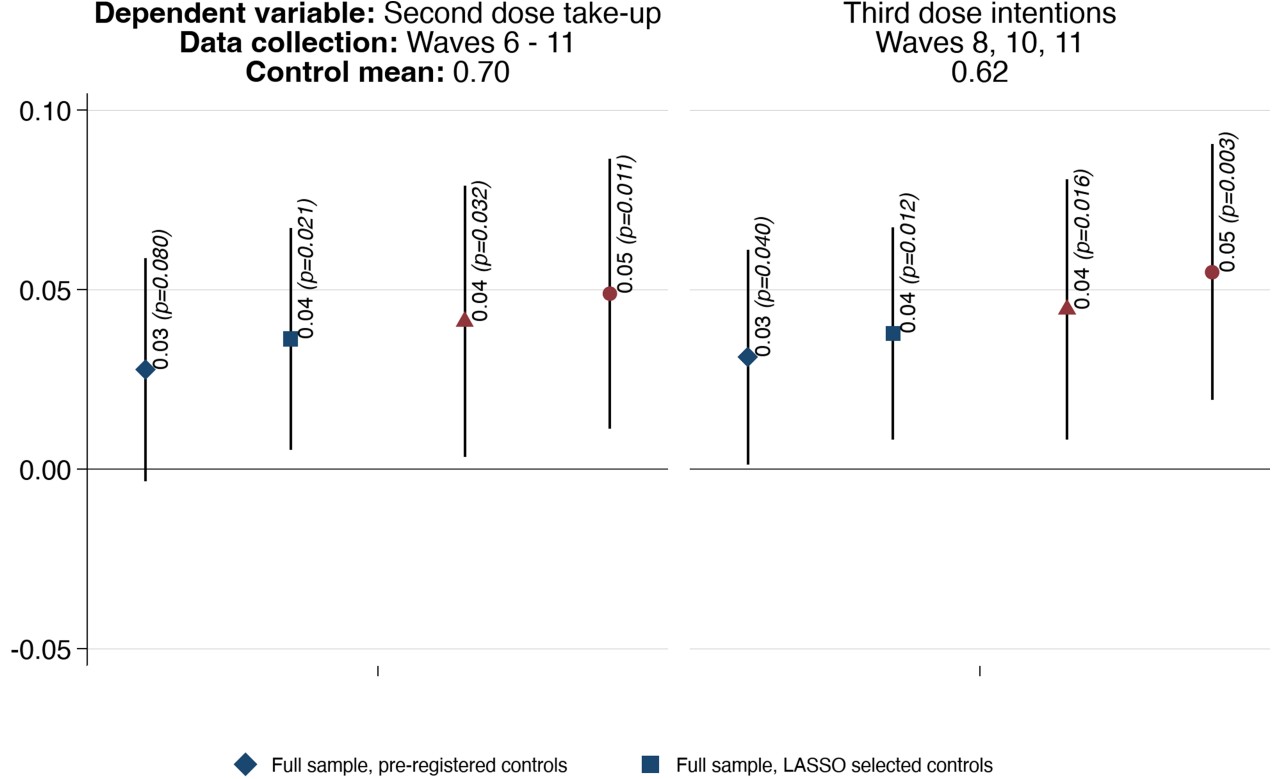

**Extended Data Fig. 2 | Effects of the Consensus condition on the second dose uptake and on intentions to uptake a third (booster) dose (Main Experiment, Sample of adult Czech population).** This figure plots estimated treatment effects on 1) the second dose uptake (two doses were designed as a complete vaccination cycle for the most commonly used vaccines), and on 2) intentions to uptake a third (booster) dose. Markers show the estimated effects, the whiskers denote the 95%-confidence interval based on standard errors clustered at the individual level. Estimated effects and t-test (two-sided) p-values are reported in the Figure. No adjustments for multiple comparisons. Diamonds and triangles report estimates from a linear probability regression that controls for the pre-registered set of control variables. Squares and circles report estimates from a double-selection LASSO linear regression (dsregress command in Stata 17) selecting from a set of covariates in Extended Data Table 1. All regressions include wave fixed effects. In the upper part of the Figure we report the timing and control mean. We report estimates for the full sample (diamonds and squares) and for a restricted sample of respondents participating in all 11 waves (triangles and circles). Full sample: Consensus condition n = 807–904, Control condition n = 800–897, depending on survey wave. Fixed sample: Consensus condition n = 614; Control condition n = 598.

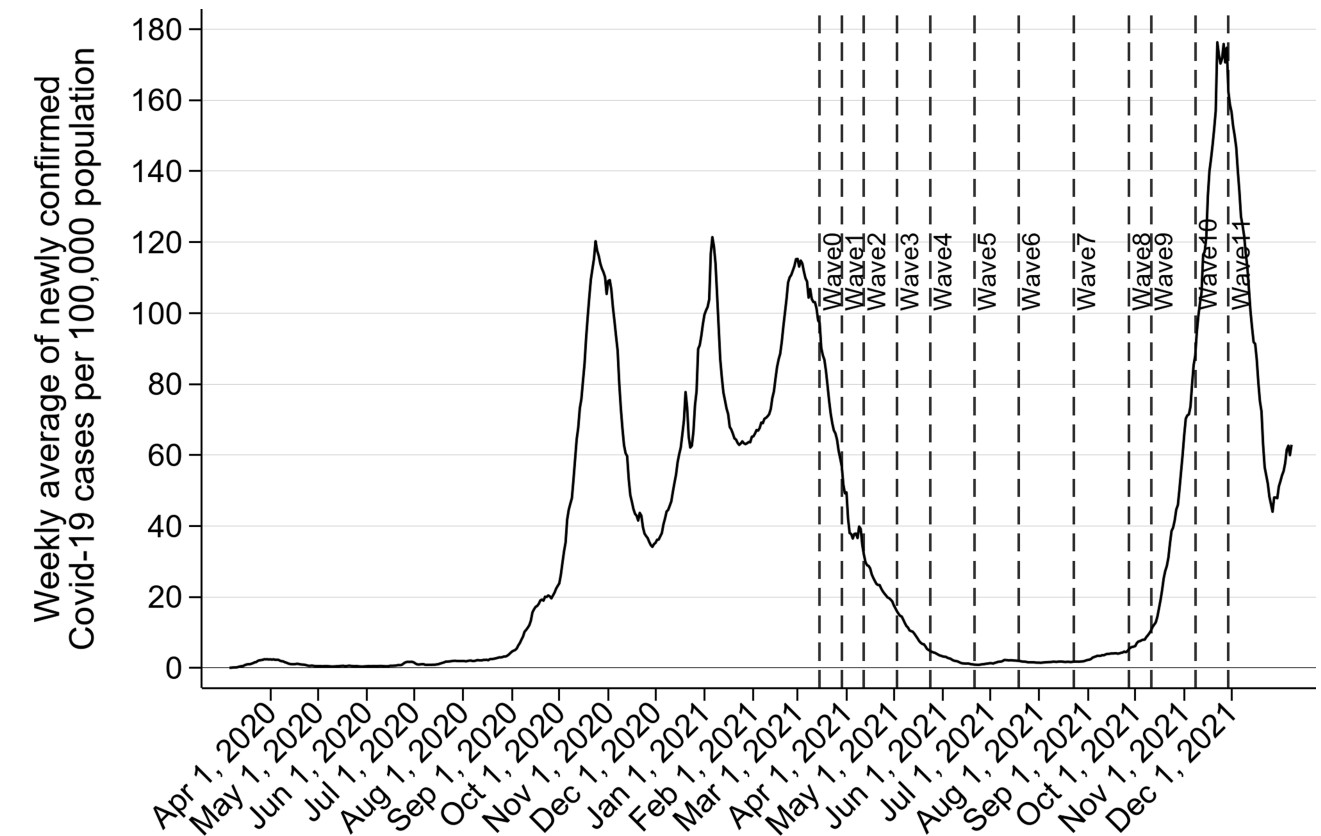

**Extended Data Fig. 3 | Weekly average of newly confirmed Covid-19 cases per 100,000 population.** Case data source: The Czech Ministry of Health (https://onemocneni-aktualne.mzcr.cz/api/v2/covid-19/osoby.csv, Accessed on January 12, 2022). Population data source: The Czech Statistical Office (https://www.czso.cz/csu/czso/obyvatelstvo-podle-petiletych-vekovych-skupin-a-pohlavi-v-krajich-a-okresech, Accessed on January 12, 2022) and ref. [38].

**Extended Data Table 1 | Demographic characteristics: summary statistics and randomization check for the full sample (Main Experiment, Sample of adult Czech population)**

| | (1) Full sample | (2) CONTROL | (3) CONSENSUS | (4) P-value | | (1) Full sample | (2) CONTROL | (3) CONSENSUS | (4) P-value |
|---|---|---|---|---|---|---|---|---|---|
| **Female** | 0.501 | 0.490 | 0.511 | 0.326 | university | 0.315 | 0.314 | 0.315 | 0.951 |
| **Age category** | | | | | **Economic status** | | | | |
| age cat 18-24 | 0.043 | 0.049 | 0.037 | 0.198 | Employee | 0.480 | 0.479 | 0.482 | 0.879 |
| age cat 25-34 | 0.114 | 0.114 | 0.113 | 0.951 | Entrepreneur | 0.046 | 0.042 | 0.050 | 0.401 |
| age cat 35-44 | 0.160 | 0.159 | 0.161 | 0.898 | Student | 0.035 | 0.038 | 0.032 | 0.480 |
| age cat 45-54 | 0.187 | 0.182 | 0.192 | 0.531 | Parental leave | 0.039 | 0.041 | 0.036 | 0.574 |
| age cat 55-64 | 0.183 | 0.186 | 0.180 | 0.743 | Retired | 0.348 | 0.349 | 0.348 | 0.940 |
| age cat 65+ | 0.314 | 0.311 | 0.316 | 0.803 | Unemployed | 0.036 | 0.034 | 0.038 | 0.637 |
| **Household size** | 2.335 | 2.310 | 2.360 | 0.281 | Other | 0.016 | 0.017 | 0.014 | 0.601 |
| **Number of children** | 0.428 | 0.424 | 0.432 | 0.538 | **Household income** | | | | |
| **Children missing** | 0.068 | 0.069 | 0.067 | 0.800 | Up to 10,000 CZK | 0.014 | 0.017 | 0.011 | 0.271 |
| **Region** | | | | | 10,001 - 15,000 CZK | 0.065 | 0.066 | 0.065 | 0.934 |
| Prague | 0.289 | 0.294 | 0.285 | 0.640 | 15,001 - 20,000 CZK | 0.095 | 0.081 | 0.109 | 0.030 |
| Central Bohemia | 0.087 | 0.074 | 0.099 | 0.043 | 20,001 - 25,000 CZK | 0.075 | 0.077 | 0.072 | 0.683 |
| South Bohemia | 0.038 | 0.043 | 0.033 | 0.256 | 25,001 - 30,000 CZK | 0.108 | 0.108 | 0.107 | 0.894 |
| Plzeň | 0.046 | 0.049 | 0.043 | 0.534 | 30,001 - 35,000 CZK | 0.123 | 0.122 | 0.124 | 0.888 |
| Karlovy Vary | 0.018 | 0.018 | 0.018 | 0.998 | 35,001 - 40,000 CZK | 0.109 | 0.123 | 0.096 | 0.051 |
| Ústí | 0.067 | 0.066 | 0.068 | 0.857 | 40,001 - 50,000 CZK | 0.122 | 0.126 | 0.118 | 0.599 |
| Liberec | 0.038 | 0.036 | 0.039 | 0.728 | 50,001 - 60,000 CZK | 0.090 | 0.076 | 0.104 | 0.027 |
| Hradec Králové | 0.042 | 0.039 | 0.045 | 0.511 | Over 60,000 CZK | 0.085 | 0.081 | 0.089 | 0.526 |
| Pardubice | 0.041 | 0.040 | 0.042 | 0.822 | I don't know / Don't want to say | 0.115 | 0.124 | 0.106 | 0.196 |
| Vysočina | 0.034 | 0.030 | 0.038 | 0.335 | **Vaccine intention (Wave -1)** | 0.642 | 0.642 | 0.641 | 0.951 |
| South Moravia | 0.097 | 0.107 | 0.088 | 0.143 | **Vaccinated** | 0.082 | 0.091 | 0.072 | 0.113 |
| Olomouc | 0.049 | 0.046 | 0.051 | 0.539 | **Beliefs about doctors'[+]** | | | | |
| Zlín | 0.045 | 0.050 | 0.040 | 0.250 | Intentions to get vaccinated | 57.163 | 58.146 | 56.180 | 0.053 |
| Moravia-Silesia | 0.110 | 0.108 | 0.111 | 0.828 | Trust in Covid-19 vaccines | 61.495 | 62.278 | 60.712 | 0.059 |
| **Town size** | | | | | **Observations** | 2,101 | 1,051 | 1,050 | |
| Below 999 | 0.065 | 0.052 | 0.077 | 0.021 | **Omnibus randomization test of joint significance for all variables above** | | | | |
| 1,000-1,999 | 0.035 | 0.031 | 0.038 | 0.402 | P-value | | | | 0.342 |
| 2,000-4,999 | 0.059 | 0.059 | 0.059 | 0.996 | | | | | |
| 5,000-19,999 | 0.111 | 0.107 | 0.116 | 0.483 | | | | | |
| 20,000-49,999 | 0.072 | 0.063 | 0.081 | 0.107 | | | | | |
| 50,000-99,999 | 0.171 | 0.182 | 0.160 | 0.186 | | | | | |
| Above 100,000 | 0.487 | 0.506 | 0.469 | 0.085 | | | | | |
| **Education** | | | | | | | | | |
| primary | 0.046 | 0.039 | 0.052 | 0.142 | | | | | |
| lower secondary | 0.277 | 0.260 | 0.293 | 0.085 | | | | | |
| upper secondary | 0.363 | 0.387 | 0.339 | 0.022 | | | | | |

Means in columns 1, 2, and 3. Column 4 reports p-values of a Wilcoxon rank-sum test (two-sided) for equality between the Control and Consensus conditions for non-binary variables (Household size, Number of children, Beliefs about doctors' intentions and Beliefs about doctors' trust), whereas for all remaining categorical variables we use Pearson's chi-squared test. Full sample used (Consensus n=1,050; Control n=1,051). The omnibus randomization test of joint significance presents a p-value of an F-test (two-sided) for an OLS regression with Consensus as a dependent variable and the set of covariates reported in the table as independent variables for the wave 0 sample. [+] We did not elicit beliefs about the third type of information provided to respondents in the Consensus condition (the willingness of doctors to recommend Covid-19 vaccines to patients), to economize on time, since we expected this type of belief to be highly correlated with the other two about doctors' views (indeed, the pairwise correlation coefficient between Wave0 beliefs about doctors' trust and vaccination intentions is r(2,099)=0.60, p<0.01).

**Extended Data Table 2 | Timeline of the Main Experiment (Sample of adult Czech population)**

| Panel A: Timing and observations | Mar 1 | Mar 15 | Mar 29 | Apr 12 | May 03 | May 24 | Jun 21 | Jul 19 | Aug 23 | Sep 27 | Oct 11 | Nov 8 | Nov 29 |
|---|---|---|---|---|---|---|---|---|---|---|---|---|---|
| Data collection start | Mar 1 | Mar 15 | Mar 29 | Apr 12 | May 03 | May 24 | Jun 21 | Jul 19 | Aug 23 | Sep 27 | Oct 11 | Nov 8 | Nov 29 |
| Wave # | -1 | 0 | 1 | 2 | 3 | 4 | 5 | 6 | 7 | 8 | 9 | 10 | 11 |
| Observations total | 1,970 | 2,101 | 1,940 | 1,939 | 1,896 | 1,860 | 1,792 | 1,620 | 1,771 | 1,745 | 1,734 | 1,611 | 1,801 |
| Observations CONTROL | 979 | 1,051 | 970 | 973 | 947 | 925 | 893 | 813 | 884 | 878 | 865 | 800 | 897 |
| Observations CONSENSUS | 991 | 1,050 | 970 | 966 | 949 | 935 | 899 | 807 | 887 | 867 | 869 | 811 | 904 |

| Panel B: Data collected | | | | | | | | | | | | | |
|---|---|---|---|---|---|---|---|---|---|---|---|---|---|
| Beliefs (trust / take-up) | | x | x | | | | | | | | | | |
| **CONSENSUS treatment** | | x | | | | | | | | | | | |
| Vaccinated | x | x | x | x | x | x | x | x | x | x | x | x | x |
| Vaccination intentions (if not vaccinated) | x | x | x | x | x | x | x | x | x | x | x | x | x |
| Booster dose intentions | | | | | | | | | | x | | x | x |
| Vaccination certificate verification | | | | | | | | | | | | | x |
| Third party verification (two weeks after Wave11) | | | | | | | | | | | | | x |

| Panel C: Vaccine registration eligibility | 70+ | | | | 55+ | 35+ | 16+ | 12+ | 12+ | 12+ | 12+ | 12+ | 12+ |
|---|---|---|---|---|---|---|---|---|---|---|---|---|---|
| | medical workers | | severely | social care workers | | | | | | | | | |
| | school employees | | chronically ill | chronically ill | academic | | | | | | | | |

In Panel A, we report dates, wave order indicators, and numbers of participants. In Panel B, we report when the Consensus treatment was implemented and which outcome variables were collected in each wave. See Supplementary Information for exact wording of questions and of the Consensus treatment. In Panel C, we report the vaccination eligibility status of groups using the information from a government run website (https://covid.gov.cz/situace/registrace-na-ockovani/casova-osa-ockovani). We report eligibility status at the start of the data collection for a respective wave. Once a group becomes eligible, it remains eligible in subsequent waves. The only group for which the eligibility was withdrawn were school employees, on March 28, 2021. More details about the development of vaccine eligibility is in the Background section of the Supplementary Information.

**Extended Data Table 3 | Effect of the Consensus condition on respondents' vaccination uptake**

| | (1) | (2) | (3) | (4) | (5) | (6) | (7) | (8) | (9) | (10) | (11) |
|---|---|---|---|---|---|---|---|---|---|---|---|
| Dependent variable | | | | | | Vaccinated | | | | | |
| Wave | 1 | 2 | 3 | 4 | 5 | 6 | 7 | 8 | 9 | 10 | 11 |
| **Panel A: Full sample** | | | | | | | | | | | |
| **Linear probability model with pre-registered set of controls** | | | | | | | | | | | |
| CONSENSUS | -0.031** | -0.004 | 0.002 | 0.009 | 0.019 | 0.040** | 0.038** | 0.037** | 0.028* | 0.038** | 0.027* |
| | (0.014) | (0.016) | (0.018) | (0.018) | (0.018) | (0.018) | (0.017) | (0.017) | (0.017) | (0.018) | (0.016) |
| | [0.033] | [0.784] | [0.904] | [0.623] | [0.291] | [0.029] | [0.033] | [0.033] | [0.093] | [0.023] | [0.088] |
| Observations | 1,940 | 1,939 | 1,896 | 1,860 | 1,792 | 1,620 | 1,771 | 1,745 | 1,734 | 1,611 | 1,801 |
| CONTROL mean | 0.152 | 0.178 | 0.339 | 0.539 | 0.667 | 0.700 | 0.727 | 0.736 | 0.747 | 0.754 | 0.771 |
| R-squared | 0.168 | 0.194 | 0.371 | 0.447 | 0.355 | 0.354 | 0.333 | 0.334 | 0.350 | 0.330 | 0.316 |
| **Double-selection LASSO linear regression** | | | | | | | | | | | |
| | -0.012 | 0.005 | 0.012 | 0.018 | 0.027 | 0.047*** | 0.046*** | 0.045*** | 0.039** | 0.049*** | 0.037** |
| | (0.010) | (0.012) | (0.016) | (0.017) | (0.018) | (0.018) | (0.017) | (0.017) | (0.017) | (0.017) | (0.016) |
| **Panel B: Fixed sample** | | | | | | | | | | | |
| **Linear probability model with pre-registered set of controls** | | | | | | | | | | | |
| CONSENSUS | -0.028 | -0.008 | -0.003 | 0.018 | 0.037* | 0.046** | 0.057*** | 0.054*** | 0.060*** | 0.053*** | 0.044** |
| | (0.019) | (0.021) | (0.023) | (0.022) | (0.022) | (0.021) | (0.020) | (0.020) | (0.020) | (0.020) | (0.019) |
| | [0.149] | [0.687] | [0.889] | [0.424] | [0.092] | [0.032] | [0.010] | [0.003] | [0.002] | [0.006] | [0.018] |
| Observations | 1,212 | 1,212 | 1,212 | 1,212 | 1,212 | 1,212 | 1,212 | 1,212 | 1,212 | 1,212 | 1,212 |
| CONTROL mean | 0.157 | 0.189 | 0.362 | 0.557 | 0.677 | 0.714 | 0.731 | 0.741 | 0.741 | 0.754 | 0.776 |
| R-squared | 0.166 | 0.197 | 0.352 | 0.439 | 0.375 | 0.378 | 0.372 | 0.359 | 0.370 | 0.363 | 0.354 |
| **Double-selection LASSO linear regression** | | | | | | | | | | | |
| | -0.012 | 0.006 | 0.008 | 0.030 | 0.045** | 0.054*** | 0.064*** | 0.062*** | 0.070*** | 0.063*** | 0.052*** |
| | (0.013) | (0.016) | (0.022) | (0.022) | (0.021) | (0.020) | (0.020) | (0.020) | (0.019) | (0.019) | (0.019) |

OLS coefficients. Huber-White robust standard errors in parentheses. Randomization inference Z-test (two-sided) p-values in square brackets (ritest command in Stata). The dependent variable in all columns is an indicator for vaccination uptake, equal to 1 if the respondent reported having obtained at least one dose of a vaccine against Covid-19. Panel A uses the full sample. Panel B uses a sample of respondents participating in all 11 waves. Columns report results for each wave separately (wave 1 in Column 1 to wave 11 in Column 11). In all columns we use the pre-registered set of controls. Estimated coefficients from a double-selection LASSO linear regression (dsregress command in Stata 17) selecting from a set of covariates in Extended Data Table 1 are reported in the bottom parts of each panel. T-test (two-sided) p-values reported as *p<0.10; **p<0.05; ***p<0.01. No adjustments for multiple comparisons.

**Extended Data Table 4 | Effect of the Consensus condition on respondents' vaccination uptake: Robustness**

| Dependent variable | (1) | (2) | (3) | (4) | (5) | (6) | (7) | (8) | (9) | (10) |
|---|---|---|---|---|---|---|---|---|---|---|
| | | | | | Vaccinated | | | | | |
| **Waves 6-11** | | | | | | | | | | |
| **Panel A: Full sample** | | | | | | | | | | |
| CONSENSUS | 0.036* | 0.037** | 0.037** | 0.037** | 0.039** | 0.036** | 0.035** | 0.042*** | 0.042*** | 0.044*** |
| | (0.019) | (0.019) | (0.019) | (0.019) | (0.018) | (0.018) | (0.015) | (0.015) | (0.015) | (0.015) |
| Observations | 10,282 | 10,282 | 10,282 | 10,282 | 10,282 | 10,282 | 10,282 | 10,282 | 10,282 | 10,282 |
| CONTROL mean | 0.740 | 0.740 | 0.740 | 0.740 | 0.740 | 0.740 | 0.740 | 0.740 | 0.740 | 0.740 |
| R-squared | 0.004 | 0.050 | 0.052 | 0.074 | 0.092 | 0.115 | 0.333 | 0.356 | 0.357 | |
| | | | | | | | | | | |
| **Panel B: Fixed sample** | | | | | | | | | | |
| CONSENSUS | 0.050** | 0.047** | 0.047** | 0.048** | 0.047** | 0.044** | 0.052*** | 0.059*** | 0.059*** | 0.061*** |
| | (0.023) | (0.023) | (0.023) | (0.023) | (0.022) | (0.022) | (0.019) | (0.018) | (0.018) | (0.018) |
| Observations | 7,272 | 7,272 | 7,272 | 7,272 | 7,272 | 7,272 | 7,272 | 7,272 | 7,272 | 7,272 |
| CONTROL mean | 0.743 | 0.743 | 0.743 | 0.743 | 0.743 | 0.743 | 0.743 | 0.743 | 0.743 | 0.743 |
| R-squared | 0.006 | 0.043 | 0.047 | 0.081 | 0.103 | 0.129 | 0.364 | 0.388 | 0.389 | |
| | | | | | | | | | | |
| **Specification** | | | | | | | | | | |
| Pre-registered set of controls | No | No | No | No | No | No | YES | No | No | No |
| Double-selection LASSO linear regression | No | No | No | No | No | No | No | No | No | YES |
| **Controls** | | | | | | | | | | LASSO selected: |
| Wave FE | Yes | Yes | Yes | Yes | Yes | Yes | Yes | Yes | Yes | No |
| Gender | No | Yes | Yes | Yes | Yes | Yes | Yes | Yes | Yes | No |
| Age | No | Yes | Yes | Yes | Yes | Yes | Yes | Yes | Yes | Yes[+] |
| Household size | No | No | Yes | Yes | Yes | Yes | Yes | Yes | Yes | No |
| Children | No | No | Yes | Yes | Yes | Yes | Yes | Yes | Yes | No |
| Region | No | No | No | Yes | Yes | Yes | Yes | Yes | Yes | No |
| Town size | No | No | No | Yes | Yes | Yes | Yes | Yes | Yes | No |
| Education | No | No | No | No | Yes | Yes | Yes | Yes | Yes | No |
| Economic status | No | No | No | No | No | Yes | Yes | Yes | Yes | No |
| Household income | No | No | No | No | No | Yes | Yes | Yes | Yes | No |
| Prior vaccination intentions | No | No | No | No | No | No | Yes | Yes | Yes | Yes |
| Prior beliefs about doctors | No | No | No | No | No | No | No | Yes | Yes | Yes |
| Prior vaccination status | No | No | No | No | No | No | No | No | Yes | Yes |

OLS coefficients. Standard errors clustered at the respondent level in parentheses. The dependent variable in all columns is an indicator for vaccination uptake, equal to 1 if the respondent reported having obtained at least one dose of a vaccine against Covid-19. Panel A uses the full sample: Consensus condition n=5,145 (981 clusters=respondents); Control n=5,137 (983 clusters=respondents). Panel B uses a sample of respondents participating in all 11 waves: Consensus n=3,684 (614 clusters=respondents); Control n=3,588 (598 clusters=respondents). We use data on the uptake from waves 6–11 when vaccines were available for all adults. Columns 1–9 report results from regressions by adding sets of controls as indicated in the bottom part of the table. The categories correspond to controls as presented in Extended Data Table 1. Column 7 uses the pre-registered set of controls. Column 10 reports results from a double-selection LASSO linear regression model (dsregress command in Stata 17) selecting from a set of covariates in Extended Data Table 1, reported in the bottom parts of each panel. The categories from which LASSO selected controls are indicated by "Yes". [+]LASSO selected age to be included among control variables for the estimates for the full sample (but not for the fixed sample). All columns include wave fixed effects. T-test (two-sided) p-values reported as *p<0.10; **p<0.05; ***p<0.01. No adjustments for multiple comparisons.

# Extended Data Table 5 | Effect of the Consensus condition on respondents' vaccination uptake: additional results

| | (1) | (2) | (3) | (4) | (5) | (6) | (7) | (8) | (9) | (10) | (11) |
|---|---|---|---|---|---|---|---|---|---|---|---|
| Dependent variable | | | | | | Vaccinated | | | | | |
| Sample | Full | Fixed | Imputation from below | Imputation from above | Attentive | Underestimating trust | Overestimating trust | Underestimating take-up | Overestimating take-up | Prior intention: Not to get vaccinated | Prior intention: To get vaccinated |
| **Waves 6-11** | | | | | | | | | | | |
| **Linear probability model with pre-registered set of controls** | | | | | | | | | | | |
| CONSENSUS | 0.035** | 0.052*** | 0.028* | 0.030* | 0.035** | 0.037** | 0.018 | 0.040** | -0.027 | 0.077** | 0.016 |
| | (0.015) | (0.019) | (0.015) | (0.015) | (0.015) | (0.017) | (0.020) | (0.016) | (0.039) | (0.038) | (0.013) |
| | [0.006] | [0.009] | [0.082] | [0.060] | [0.030] | [0.044] | [0.427] | [0.016] | [0.594] | [0.050] | [0.013] |
| Wave FEs | Yes | Yes | Yes | Yes | Yes | Yes | Yes | Yes | Yes | Yes | Yes |
| Observations | 10,282 | 7,272 | 12,606 | 11,361 | 9,900 | 9,061 | 1,221 | 9,598 | 684 | 3,506 | 6,776 |
| CONTROL mean | 0.740 | 0.743 | 0.688 | 0.740 | 0.743 | 0.708 | 0.971 | 0.724 | 0.948 | 0.393 | 0.920 |
| R-squared | 0.333 | 0.364 | 0.347 | 0.327 | 0.341 | 0.321 | 0.430 | 0.338 | 0.471 | 0.074 | 0.078 |
| Comparison chi-sq (p-value) | | | | | | 0.55 (0.458) | | 2.67 (0.102) | | 11.37 (0.001) | |
| **Double-selection LASSO linear regression** | | | | | | | | | | | |
| CONSENSUS | 0.044*** | 0.061*** | 0.038** | 0.040*** | 0.044*** | 0.048*** | 0.007 | 0.048*** | -0.068 | 0.090** | 0.020 |
| | (0.015) | (0.018) | (0.015) | (0.015) | (0.015) | (0.017) | (0.020) | (0.016) | (0.046) | (0.036) | (0.014) |
| Comparison chi-sq (p-value) | | | | | | 2.37 (0.124) | | 5.66 (0.017) | | 3.32 (0.069) | |

OLS coefficients. Standard errors clustered at the respondent level in parentheses. Randomization inference p-values in square brackets (ritest command in Stata). The dependent variable in all columns is an indicator for vaccination uptake, equal to 1 if the respondent reported having obtained at least one dose of a vaccine against Covid-19. Wave 6–11 sample used. Column 1 uses the full sample: Consensus condition n=5,145 (981 clusters=respondents); Control n=5,137 (983 clusters=respondents). Column 2 uses a sample of respondents participating in all 11 waves: Consensus n=3,684 (614 clusters=respondents); Control n=3,588 (598 clusters=respondents). Column 3 imputes missing vaccination uptake data by using the latest vaccination status in an earlier wave for each missing wave. Column 4 imputes missing vaccination uptake data by using the first reported vaccination status in a non-missing subsequent wave. Column 5 restricts the full sample to respondents who passed all attention checks embedded in the survey. Columns 6 and 7 restrict the sample to respondents underestimating and overestimating trust in the Covid-19 vaccines, respectively. Columns 8 and 9 restrict the sample to respondents underestimating and overestimating doctors' intentions to get vaccinated, respectively. Columns 10 and 11 restrict the sample to respondents without and with intentions to get vaccinated prior to wave 0, respectively. In all columns we use the pre-registered set of controls. All columns include wave fixed effects. Estimated coefficients from a double-selection LASSO linear regression (dsregress command in Stata 17) selecting from a set of covariates in Extended Data Table 1 are reported in the bottom part of the panel. Rows titled "Comparison" in each panel report a chi-square statistic and a p-value for a test of equivalence of coefficients across two respective models estimated using seemingly unrelated regressions (suest command in Stata 17). For LASSO selected controls, we use OLS models with controls selected by LASSO. T-test (two-sided) p-values reported as *p<0.10; **p<0.05; ***p<0.01. No adjustments for multiple comparisons.

## Extended Data Table 6 | Third party and certificate verification

| | (1) | (2) | (3) | (4) | (5) | (6) | (7) | (8) | (9) | (10) |
|---|---|---|---|---|---|---|---|---|---|---|
| | Observations | Response rate relative to Wave11 sample | | | Verification rate for self-reported vaccinated | | | Verification rate for self-reported unvaccinated | | |
| | | CONSENSUS | CONTROL | chi-sq (p-value) | CONSENSUS | CONTROL | chi-sq (p-value) | CONSENSUS | CONTROL | chi-sq (p-value) |
| **Panel A: Third party verification** | 1672 | 0.936 | 0.921 | 1.52 (0.217) | 0.999 | 0.998 | 0.002 (0.967) | 0.964 | 0.935 | 1.52 (0.218) |
| **Panel B: Certificate verification** | 1364 | 0.960 | 0.970 | 1.00 (0.318) | 0.941 | 0.949 | 0.473 (0.492) | | | |

Column 1 reports observations for those who participated in wave 11 and [Panel A: participated in the third party verification, Panel B: reported being vaccinated with at least one dose of a vaccine against Covid-19]. Sample means in columns 2, 3, 5, 6, 8, 9. Columns 4, 7, and 10 report Pearson's chi-squared test F-statistic and a corresponding p-value in parentheses. Supplementary Section 3.4 describes both verification methods.

# Reporting Summary

## Statistics

For all statistical analyses, confirm that the following items are present in the figure legend, table legend, main text, or Methods section.

| n/a | Confirmed | |
|---|---|---|
| ☐ | ☒ | The exact sample size ($n$) for each experimental group/condition, given as a discrete number and unit of measurement |
| ☐ | ☒ | A statement on whether measurements were taken from distinct samples or whether the same sample was measured repeatedly |
| ☐ | ☒ | The statistical test(s) used AND whether they are one- or two-sided<br>*Only common tests should be described solely by name; describe more complex techniques in the Methods section.* |
| ☐ | ☒ | A description of all covariates tested |
| ☐ | ☒ | A description of any assumptions or corrections, such as tests of normality and adjustment for multiple comparisons |
| ☐ | ☒ | A full description of the statistical parameters including central tendency (e.g. means) or other basic estimates (e.g. regression coefficient) AND variation (e.g. standard deviation) or associated estimates of uncertainty (e.g. confidence intervals) |
| ☐ | ☒ | For null hypothesis testing, the test statistic (e.g. $F$, $t$, $r$) with confidence intervals, effect sizes, degrees of freedom and $P$ value noted<br>*Give P values as exact values whenever suitable.* |
| ☒ | ☐ | For Bayesian analysis, information on the choice of priors and Markov chain Monte Carlo settings |
| ☒ | ☐ | For hierarchical and complex designs, identification of the appropriate level for tests and full reporting of outcomes |
| ☐ | ☒ | Estimates of effect sizes (e.g. Cohen's $d$, Pearson's $r$), indicating how they were calculated |

*Our web collection on statistics for biologists contains articles on many of the points above.*

## Software and code

Policy information about availability of computer code

| Data collection | Main study / third party verification: CAWI by NMS Market Research (software: NMS CAWI) / STEM/MARK (software iQuest); Survey among medical doctors: Qualtrics; survey distributed by the Czech Medical Chamber |
|---|---|
| Data analysis | Stata 17; Non-native Stata packages used: ritest (version 2017), orth_out (version 2016), specc (version 2019), grc1leg (version 2014) |

For manuscripts utilizing custom algorithms or software that are central to the research but not yet described in published literature, software must be made available to editors and reviewers. We strongly encourage code deposition in a community repository (e.g. GitHub). See the Nature Portfolio guidelines for submitting code & software for further information.

## Data

Policy information about availability of data

All manuscripts must include a data availability statement. This statement should provide the following information, where applicable:

- Accession codes, unique identifiers, or web links for publicly available datasets
- A description of any restrictions on data availability
- For clinical datasets or third party data, please ensure that the statement adheres to our policy

The datasets generated during and analyzed during the current study, together with replication files, is available in the Harvard Dataverse repository: https://doi.org/10.7910/DVN/RH0T6R

The availability of the dataset from the Supplementary survey with medical doctors is subject to approval of the Czech Medical Chamber upon request. The reason for not publishing the data is that the doctors were informed that their responses will only be published in an aggregated form to ensure full anonymity. We were aware that some doctors with unique characteristics could potentially be identified from an anonymized but individual-level dataset. Aggregated data can be provided to researchers upon request, additional analysis on an individual level could be requested from the authors.

# Field-specific reporting

Please select the one below that is the best fit for your research. If you are not sure, read the appropriate sections before making your selection.

☐ Life sciences ☒ Behavioural & social sciences ☐ Ecological, evolutionary & environmental sciences

For a reference copy of the document with all sections, see nature.com/documents/nr-reporting-summary-flat.pdf

# Behavioural & social sciences study design

All studies must disclose on these points even when the disclosure is negative.

| | |
|---|---|
| Study description | Online experiment (quantitative experimental data) on the effect of providing information about medical doctors' consensus on vaccination demand and vaccination take-up. Accompanying survey among Czech medical doctors provides data for the information treatment in the online experiment. |
| Research sample | Main experiment (third party verification drawn from the same subject pool as main experiment): Representative sample of the Czech population 18+ in terms of sex, age, education, region, municipality size, employment status before the Covid-19 pandemic, age x sex, and age x education (n = 2,101), sampled from the online panel called "Český národní panel" (Czech national panel). Prague and municipalities above 50,000 are oversampled (boost 200%). The respondents are part of a high-frequency longitudinal study "Life during the pandemic". This allowed us to naturally implement the information intervention and to continue asking questions on vaccination intentions and take-up in a setting familiar to the respondents. The fact that the participants have been participating in up to 24 previous waves of the study prior to the intervention allows us to maintain low rates of attrition and to measure key outcomes repeatedly over an extended period of time. Survey among medical doctors (n = 9,650): survey distributed to all members of the Czech Medical Chamber, response rate 24%. Representativeness discussed in Supplementary Table 1. |
| Sampling strategy | Main experiment (third party verification drawn from the same subject pool as main experiment): Quota sampling (based on the characteristics specified above) from an online panel called "Český národní panel" (Czech national panel). There are above 1,000 participants in each of the two experimental conditions and thus we are powered to detect even relatively small effects. For vaccination intentions/vaccination take-up in the control group between 0.15-0.75, having a sample size of 1,000 respondents per condition allows us to detect an effect of 0.042-0.056 (alpha=0.05, power=80%, one-sided). Survey among medical doctors: survey distributed to all members of the Czech Medical Chamber |
| Data collection | Main experiment, third party verification, and Survey among medical doctors: Participants complete the study on their computers or smartphones. |
| Timing | Main experiment: Wave0: March 15, Wave1: March 29, Wave2: April 12, Wave3: May 3, Wave4: May 24, Wave5: June 21, Wave6: July 19, Wave7: August 23, Wave 8: September 27, Wave 9: October 11, Wave 10: November 8, Wave 11: November 29 (All data collected in 2021; Dates represent the first day of data collection; data collection typically ensued over 5 consecutive days); Survey among medical doctors: February 11-24, 2021; Third party verification: December 15-23, 2021 |
| Data exclusions | Main experiment: we use data for all Wave0 participants (n=2,101); we also report results for a "fixed sample" of participants participating in all eleven waves (n=1,212). Survey among medical doctors: 11,655 respondents opened the survey. Of these, 1,164 answered that they do not currently work in healthcare, 83 workers in healthcare answered that they are not medical doctors and 92 answered that they do not work in the Czech Republic. We excluded these respondents from the analysis. 666 respondents did not complete the survey. In the analysis, we work with sample of 9,650 Czech medical doctors who completed the survey. Third party verification: we use all data for main experiment Wave11 participants (n=1,672). We excluded 50 participants who participated in the Third party verification but not in main experiment Wave11. |
| Non-participation | Main experiment: The response rate, as compared to the base sample from Wave0, was 92% in Wave1 (March), 92% in Wave2 (April), 90% in Wave3 (May), 89% in Wave4 (May), 85% in Wave5 (June), 77% in Wave6 (July), 84% in Wave7 (August), 83% in Wave 8 (September), 82% in Wave 9 (October), 76% in Wave 10 (early November), and 86% in Wave 11 (late November). Survey among medical doctors: see data exclusions. 666 respondents who satisfy criteria did not complete the survey. Response rate of 24%. Third party verification: 7% of main experiment Wave11 respondents did not participate. |
| Randomization | Main experiment: Random allocation of participants into CONTROL and CONSENSUS conditions by a computer algorithm. No randomization in the Survey among medical doctors. |

# Reporting for specific materials, systems and methods

We require information from authors about some types of materials, experimental systems and methods used in many studies. Here, indicate whether each material, system or method listed is relevant to your study. If you are not sure if a list item applies to your research, read the appropriate section before selecting a response.

## Materials & experimental systems

| n/a | Involved in the study |
|-----|----------------------|
| ☒ ☐ | Antibodies |
| ☒ ☐ | Eukaryotic cell lines |
| ☒ ☐ | Palaeontology and archaeology |
| ☒ ☐ | Animals and other organisms |
| ☐ ☒ | Human research participants |
| ☒ ☐ | Clinical data |
| ☒ ☐ | Dual use research of concern |

## Methods

| n/a | Involved in the study |
|-----|----------------------|
| ☒ ☐ | ChIP-seq |
| ☒ ☐ | Flow cytometry |
| ☒ ☐ | MRI-based neuroimaging |

# Human research participants

Policy information about studies involving human research participants

| | |
|---|---|
| Population characteristics | See above, Extended Data Table 1 and Supplementary Tables 1 and 3. |
| Recruitment | Main experiment (third party verification drawn from the same subject pool as main experiment): Members of an online panel "Český národní panel" (Czech national panel) were invited to participate in a survey. Respondents for the panel are recruited by phone calls to randomly generated phone numbers, their identity is cross-validated, they are motivated by financial and non-financial incentives. Internet access is a pre-requisite for participation in our study. Participants were randomized into the experimental conditions. Survey among Czech medical doctors: Members of the Czech Medical Chamber who opted for electronic communication (70%) invited to participate in an online survey. Membership in CMC is mandatory for all Czech medical doctors by law. |
| Ethics oversight | The research was approved by the Commission for Ethics in Research of the Faculty of Social Sciences of Charles University. |

Note that full information on the approval of the study protocol must also be provided in the manuscript.

