## [Peer Review File · Nature]

Manuscript Title: Communicating doctors' consensus persistently increases Covid-19 vaccinations

Reviewer Comments & Author Rebuttals

Reviewer Reports on the Initial Version:

Referees' comments:

Referee #1 (Remarks to the Author):

This is a RCT in the Czech Republic of an intervention designed to increase COVID-19 vaccine willingness and take-up by correcting misperceptions in the fraction of the country's doctors that believe the vaccine is safe and effective. The novelty is not in the topic or approach- but I haven't recently seen one of this scale. The results are well done statistically and the conclusions seem valid with the caveats below. Overall, this is a very nice study.

Comments for the authors to address:

The authors have a couple sentences about demand bias right before the conclusion. Obviously this is the main threat to this analysis.

For the issue of demand bias: can the authors collect independently verifiable data to suggest people actually did get vaccinated? Ideally they would pair with administrative data or have another entity ask about vaccination status. Despite my concerns, I do agree that the persistence of the results is not consistent with priming (but doesn't wholly rule out demand). Instead of relegating this to an outlet where this paper would receive less exposure, I would like to see admin data or a third-party verify these results. And if that's not possible a more robust discussion of the study limitation in the spirit of transparency and science.

One spec should be LASSO chosen controls – since they had to deviate from the pre-analysis plan.

It was not clear how representative these doctors were. Supp table 1 is not exhaustive of population-level doctor characteristics and doesn't include p-values – there's also formatting issues. This could imply there is an element of deception in the "CONSENSUS" treatment if it's only picking up one part of the doctor distribution. Then in fact, the citizens are actually correct about the "true" trust in the vaccination across all Czech doctors. The need to address this.

The magnitudes appear quite small in percent terms. The discussion should talk more about these magnitudes.

The survey doesn't include the script shown to the actual citizens (verbatim) about the CONSENSUS treatment.

Referee #2 (Remarks to the Author):

This paper examines a mismatch between doctors' beliefs about COVID-19 vaccines and the public's perceptions of those beliefs. This mismatch is then addressed by providing people with accurate information about doctors' beliefs. According to the analyses and discussion in this paper, this increases intentions to get vaccinated and actual subsequent (self-reported) vaccine take-up, even weeks later.

I quite liked the approach in this paper, including the collection of data about doctors' beliefs and then applying it in this survey. I hope that this has already encouraged propagating this information about doctors' beliefs more widely, at least in the Czech Republic, where this study was conducted.

The point estimates of the effects are quite large. As a comparison point, I thought of work that instead invokes other "everyday people" as the reference group (Moehring et al. 2021, Santos et al. 2021). The former has a similar design, but finds significant though much smaller effects. The latter has a different setup (in that it uses text message reminder and short-term appointment signup as the outcome) and finds perhaps similarly sized effects. Anyway, the authors might think about highlighting that it seems like their "expert consensus" intervention is more effective than other "social norms" interventions. As someone who has worked on the latter, that is one thing that struck me in reading this. (Though also perhaps the present work provides only limited evidence about effect sizes, given the substantial uncertainty in the estimates.)

Overall, I generally had a quite positive initial impression of this paper. And I think the whole thing makes intuitive and theoretical sense.

But I did have some concerns as I started digging into the details. These led me to wonder about the strength of the statistical evidence here. I elaborate on this below, but it seems important to better explore the robustness of the results to which covariates are adjusted for (particularly including the pre-registered analysis) and to probe questions about attrition.

Strength of evidence, imbalance, and adjustment for covariates

Contrary to the pre-analysis plan, the main analyses include adjustment for some additional covariates:

"a non-pre-specified variable for being vaccinated in Wave0 and Wave0 beliefs about the views of doctors. We added the non-specified variables due to a detected imbalance in randomization." (SI p. 32)

These indeed seem like relevant covariates to adjust for. However, this kind of data-contingent adjustment is potentially worrying. If there were indeed a problem with randomization, one would want to get to the bottom of that. But I don't see much evidence than anything was wrong; it is simply the case that there is a marginally significant imbalance ($.05 < p < .1$) in two covariates and a non-significant ($p > .1$) imbalance in another — without any correction for multiple hypothesis testing. This kind of data-contingent adjustment can increase error rates (e.g., Mutz et al. 2019), especially if no particular rule is followed, creating a "garden of forking paths" (Gelman & Loken 2014). Thus, unless the authors actually think randomization did not occur as planned (in which case perhaps more investigation is needed), I don't see why these variables should be adjusted for in all

main analyses. (Note also that there is no single obvious way to adjust for these covariates. The beliefs about doctors are often discussed in a dichotomous way, e.g., “Underestimating” vs “Overestimating” trust so one could imagine the adjustment being for that dichotomized version additionally or instead. This helps to create many possible specifications, and only one is reported.)

I also find it a bit strange that the main text says:

“Supplementary Table 3 provides evidence that the range of individual characteristics, including gender, age, household income, employment status, education, region and size of municipality, as well as vaccination intentions measured three weeks before the intervention are balanced across the experimental condition. The exceptions are ‘own vaccination’ and ‘prior beliefs about opinions of the doctors’ (measured in Wave0). Prior to the intervention, compared to participants in the CONTROL condition, the individuals in the CONSENSUS condition were less likely to be vaccinated themselves, and expected a smaller percentage of doctors to trust the vaccine and to intend to get vaccinated.” (p. 5)

Looking at Table 3, it is a bit hard to see by what definition vaccination and prior beliefs about doctors are imbalanced but we don’t have imbalance in education and region (which have tests with $p < 0.1$). And it is also kind of hard to see how there is evidence of imbalance in own vaccination; if I am reading Table S3 correctly, the associated p-value here is 0.113 (i.e., not even $p < 0.1$). Now I guess this wouldn’t matter for creating further adjustment sets, since these variables are already in the preregistered set, but it does seem that this summary in the main text is at least confusing. Perhaps the authors have in mind some correction for the multiple categorical levels of education and region? More generally, I would suggest reporting a joint test of all of these covariates being randomized; presumably this retains the null. (Furthermore, it is far from clear to be that there is any real evidence of imbalance in these additional covariates, particularly conditional on everything else being adjusted for already.)

As far as I can tell, the only variation on this analysis offered is one without controls altogether (for which the estimates is not statistically significant at $p < 0.05$, whether for vaccine demand in wave 1, Table S9, or vaccine takeup in wave 7, Table S11). That is, we don’t ever get the preregistered analysis. This seems like an important omission, which should be corrected. This absence also seems at odds with the claim in the main text that “Second, the effects on vaccination demand and vaccine take-up are robust to changes in the set of control variables.” Unless I am misunderstanding, again this only is true in the sense that there are some analyses entirely without covariates (e.g., column 2 of Table S11).

Overall, this made me substantially less confident in the main results than I was initially. Perhaps this can be addressed by adding these additional analysis; I can’t know. [I would also mention that, given the relatively large rank of the covariates, finite-sample bias may not be so small and so the authors might consider using an adjustment method with lower bias, as in Lin (2013) and Negi & Wooldridge (2021).]

Attrition

There is some attrition from the survey: “The response rate, as compared to the 187 base sample from Wave0, was 92% in Wave1 (March), 92% in Wave2 (April), 90% in 188 Wave3 (May), 89% in Wave4 (May), 85% in Wave5 (June), 77% in Wave6 (July) and 84% in

189 Wave7 (August).” The authors note that the overall rate does not differ between treatment arms.

I think this analysis could be made more thorough. In particular, different average rates of attrition are not the only means by which attrition can bias results; differential attrition of different types of respondents is sufficient. So I would suggest conducting tests of differential attrition by baseline covariates, say, by a regression as in Table A4, but with interactions with covariates. A likely less powerful test could look at balance within the non-attrited sample. (In either case, this should be done so as to avoid problems of multiple comparisons, such as via a joint test.)

More generally, the authors might consider conducting some kind of sensitivity analysis or bounding exercise here to highlight how much of an issue this would all need to be to substantially alter the results.

Heterogeneous effects

There is some discussion of heterogeneity of the effects. While it may be reasonable to say that the effects are “concentrated” in some particular subgroups, I think some of the discussion here commits the fallacy of comparing a statistically significant estimate and a non-significant one and assuming that this comparison is itself statistically significant. To provide evidence of heterogeneity per se, one should test that the effects in these subgroups are different. This can be done in a combined model with a Wald test. Or if the estimates are derived from entirely separate sets of data, as I believe all these analyses are, it can be done simply using the estimates and standard errors.

I did this for vaccine take-up for the subgroups formed by prior intention for Wave 7 (Table S11, Panel B, columns 9 and 10). The associated z-statistic is $(.090 - .023) / \sqrt{.041^2 + .016^2} = 1.52$. That is, the estimated effects in these two subgroups are not distinguishable. Eyeballing the rest of these, in Table S11 (which is the primary table supporting the claim in the main text, which is about take-up), I don’t think any are significant.

Tables S13 and S14 present analyses disaggregated by many different covariates. These are described as analyses of heterogeneous effects; however, no tests of heterogeneity per se are conducted. Rather these are just tests of a number of conditional average treatment effects within each subgroup. So my comments above apply. I would also suggest that perhaps these numerous subgroup analyses (including those chosen to be highlighted in the main text) should somehow either (a) be subject to adjustment for multiple comparisons, (b) subsumed into some more flexible procedure for finding heterogeneous effects (e.g., causal forests), and/or (c) flagged as post hoc (particularly the ones in the main text).

Other comments:

I might suggest the authors use a term other than “experimental manipulation” throughout, particularly given the broad readership for this kind of work, as it can give a wrong, nefarious-sounding impression.

Some places (e.g., Figure 4) round p-values to 0.00. I would suggest they should instead either be stated as inequalities, using more digits, or using scientific notation. The latter, in particular, allows

readers to more readily assess the strength of the evidence.

References

Gelman, A., & Loken, E. (2014). The Statistical Crisis in Science. *American Scientist*, 102(6), 460.

Lin, W. (2013). Agnostic notes on regression adjustments to experimental data: Reexamining Freedman's critique. *The Annals of Applied Statistics*, 7(1), 295-318.

Moehring, A., Collis, A., Garimella, K., Rahimian, M. A., Aral, S., & Eckles, D. (2021). Surfacing norms to increase vaccine acceptance. Available at SSRN 3782082.

Mutz, D. C., Pemantle, R., & Pham, P. (2019). The perils of balance testing in experimental design: Messy analyses of clean data. *The American Statistician*, 73(1), 32-42.

Negi, A., & Wooldridge, J. M. (2021). Revisiting regression adjustment in experiments with heterogeneous treatment effects. *Econometric Reviews*, 40(5), 504-534.

Santos, H. C., Goren, A., Chabris, C. F., & Meyer, M. N. (2021). Effect of targeted behavioral science messages on COVID-19 vaccination registration among employees of a large health system: A randomized trial. *JAMA Network Open*, 4(7), e2118702-e2118702.

Referee #3 (Remarks to the Author):

The manuscript presents the findings from an experiment that tested the impact of a behavioral intervention -- informing people about the percent of doctors that support/trust COVID-19 vaccination -- on vaccination uptake in the Czech Republic.

The information "nudge" was used because, as the authors describe, residents of the Czech Republic seemed to have an incorrect view, underestimating the percent of doctors that support/trust COVID-19 vaccination.

Some of the great features of the study:

- It is a field experiment, randomized control trial,
- studying a non-US/UK sample,
- measuring behavior (vaccination uptake), and
- "longitudinal" -- measuring the effect of the intervention over a period of roughly 5 months.

I think this paper is important as we do not have many experimental findings about interventions that measure impact on COVID-19 uptake. However, the paper also has a few limitations that give me pause and that at minimum need to be made clearer to the readers of this paper as they assess the learnings / take-aways from this study:

1. We only seem to have participants' self-reported vaccination status.
2. There seems to be an error of randomization on key variables. Specifically, the control and treatment group differ at baseline, that is, prior to intervention, in terms of
 - their beliefs about doctors' opinions (trust and intentions to get vaccinated were significantly lower in the treatment group) and
 - the participants' own vaccination rates (lower in treatment group).

While the authors control for these imbalances, this aspect gives me pause because (a) vaccination rates is also the key dependent variable later on, and (b) (mis)beliefs about doctors' opinions is supposedly the focal issue that drives vaccination rates.

In addition, the authors only consider that "these baseline differences could potentially contribute to underestimation of the treatment effects." But what they do not consider is the possibility that because of this a-priori difference – even without an intervention – the participants in the CONSENSUS condition were more likely to catch up with the remaining population as more people became eligible for vaccination, likely knew more people that got it, and more information about the vaccines became widely known.

Particularly, given that the authors also point out "It is important to bear in mind that not everyone was able to get vaccinated from the very beginning of the data collection period. Different demographic groups became eligible to register for the vaccine at different points in time.", it makes me wonder, if there are other reasons (other than the information nudge) that could explain why the CONSENSUS condition participants ended up having higher vaccination rates than those in the CONTROL group.

3. It is puzzling to me that we only see significant effects on the key dependent variable vaccination uptake in the last 2-3 surveys. That is, if I understand right, the intervention is delivered in wave 0 once, and only after three to five months do we see an effect of that one-time intervention. The authors do not comment on that enough and try to explain why that would be the case.

4. Even if the participants represent a representative sample of the Czech Republic, how likely is the informational nudge intervention to be successful if rolled out to the entire population? I am wondering about that because the experiment was run in a special setting: a sample of individuals who agreed to be part of a panel and be interviewed regularly. In addition, it is not clear to me, how could the intervention tested in this manuscript be rolled out to the general population such that we could assume it would receive the same amount of attention as the panel participants devoted to the delivered survey? For example, if one were to send this the informational nudge about doctor's beliefs via informational mail-in flyers from the government or media ads, could we just assume that that would be successful given the experiment described in this manuscript? I am doubtful.

Together, while I admire the authors' efforts and do think it is important that we gather more information on the effectiveness of easy-to-scale behavioral interventions for vaccination uptake, I am not sure how much we can learn about whether the observed effects described here are truly due to the one-time intervention and could we expected if scaled (and not clear how one could scale it).

Author Rebuttals to Initial Comments:

Dear Referees,

Thank you very much for your positive words, thorough reading of the paper, and your excellent suggestions. We have sought to carefully address every issue raised in the review. Here is a brief summary of the main improvements we have made:

- First, to address the concern about demand effects and misreporting, we collected two types of new data to verify the participant reports about their vaccination status – one set collected by an independent entity and another one aiming to link the reports with a proof of vaccination issued by the Ministry of Health of the Czech Republic. Both verifications reveal that participants misreporting their vaccine status is very rare and not related to the treatment.
- Second, we managed to obtain additional data from the Institute of Health Information and Statistics (UZIS) about the characteristics of all doctors in the county, together with information about their vaccination rates. This allows us to more convincingly document that the characteristics of doctors in our sample are very similar to the overall population of doctors in the country, and that a vast majority of doctors support vaccination, in line with information provided in the CONSENSUS condition.
- Third, we show that our estimates of the effect of the CONSENSUS condition on vaccine take-up are robust to various specifications, including the use of controls selected by the LASSO procedure, as you suggested. When we report our results, LASSO is now one of the main specifications we use. In addition, we show robustness to using the pre-specified set of control variables, difference-in-difference specification, and a host of other specifications.

In addition, we collected new data in several additional follow-up waves covering the period from September to the end of November, 2021. These unique data allow us to study the persistence of the effect in the longer-run. We find that the estimated treatment effect is remarkably stable. In the new waves, we also collected data about participant's intentions to get a (third) booster dose. The estimated effect is very similar in magnitude as the effect on take-up of the first dose. Thus, we are now able to show that the information intervention elevates vaccination demand even nine months after it was implemented (Figure 4). Further, since the vaccination demand in the CONTROL condition does not catch up with the CONSENSUS condition over such a long period, the results suggest that the type of vaccine hesitancy reduced by the CONSENSUS condition is resilient to campaigns or any life disruptions that participants were exposed to during the period studied, including the severe Covid-19 wave that took place in November 2021 in the Czech Republic, which resulted in one of the highest national mortality rates in global comparisons.

Below, we provide more detailed, point-by-point responses to all of your comments, together with specific references to what we changed in the manuscript and SI. Our responses are in black font. Your original comments are in italics and blue font.

We hope that you share our sentiment that the revisions in response to your feedback have significantly improved the paper and strengthened the evidence to make it relevant for the audience of *Nature*.

This is a RCT in the Czech Republic of an intervention designed to increase COVID-19 vaccine willingness and take-up by correcting misperceptions in the fraction of the country's doctors that believe the vaccine is safe and effective. The novelty is not in the topic or

approach- but I haven't recently seen one of this scale. The results are well done statistically and the conclusions seem valid with the caveats below. Overall, this is a very nice study.

Comments for the authors to address:

1. The authors have a couple sentences about demand bias right before the conclusion. Obviously this is the main threat to this analysis.

For the issue of demand bias: can the authors collect independently verifiable data to suggest people actually did get vaccinated? Ideally they would pair with administrative data or have another entity ask about vaccination status. Despite my concerns, I do agree that the persistence of the results is not consistent with priming (but doesn't wholly rule out demand). Instead of relegating this to an outlet where this paper would receive less exposure, I would like to see admin data or a third-party verify these results. And if that's not possible a more robust discussion of the study limitation in the spirit of transparency and science.

RESPONSE

Thank you for this point. Given that vaccination status in our data is indeed self-reported, we agree it is useful to address in much greater depth the concern that the main effect could potentially arise due to the experimenter demand motivating some participants in the CONSENSUS condition to misreport being vaccinated even when they were not. We collected two types of new data and use two approaches to address the issue.

The first approach is guided by your suggestion and recent experiments facing similar issue in different contexts¹. We use data from another, independent entity which asked the same individuals about their vaccination status in a different survey, implemented in December 2021. In the manuscript, we refer to this new data collection as third-party verification.

New data. We take advantage of the fact that different survey agencies have access to the panel our respondents are sampled from (the Czech National Panel). While the main data collection was implemented by one agency (NMS), we partnered with another agency (STEM/MARK) to include a question on vaccination status in a survey implemented on its behalf among the same sample. Since the survey agency, graphical interface, and topic of the survey were different from our main data collection, we believe respondents considered the two surveys to be completely independent of each other, and thus the experimenter demand effect potentially associated with our main survey is unlikely to affect responses in the third-party verification survey. Out of 1,801 participants who took part in the last wave of the main data collection at the end November, 1,672 also took part in the third-party verification survey, implemented two weeks later. This allows us to compare reported vaccination status at the individual level in the third-party verification survey and in our main survey for a vast majority of the sample.

Findings. We find several clear and reassuring patterns. First, mismatches in reporting are very rare in general and are not higher in the CONSENSUS condition. Only two respondents (one in CONSENSUS and one in CONTROL) reported being vaccinated in the main survey but reported the opposite in the third-party verification survey (Supplementary Table 14; see below). In addition, 18 respondents (1%) reported being vaccinated in the verification survey but not in our survey. Note that the latter type of inconsistency does not necessarily imply misreporting, because the verification survey took place two weeks after the main survey and the respondents could have been vaccinated in the meantime. In any case, this type of mismatch in reporting is also not higher in the treatment than in the control condition.

Second, we show that the treatment effect is *not* driven by participants whose reports we are unable to verify because they did not take part either in the last wave of the main data collection or in the third-party verification survey. Specifically, using ordered and multinomial logit, we show that the effect of the CONSENSUS condition on lower prevalence of participants reporting not being vaccinated is almost fully explained by greater prevalence of those reporting being vaccinated, and having their vaccination status verified in the third-party verification survey (Supplementary Table 15; see below).

The second approach to verify vaccination status aims to link reported vaccination status with official proofs of vaccination. Because we are unable to match the survey data with administrative data (which we have found is not feasible due to legal reasons, despite our efforts to pursue this direction), we decided to take advantage of the fact that all vaccinated people receive an EU Digital COVID certificate issued by the Czech Ministry of Health, and asked the respondents from our sample to provide specific pieces of information from their certificates. Importantly, the certificate contains information about the applied vaccine which is unlikely to be known by someone without a certificate. Also, respondents should usually have the document readily available, typically in a mobile app, because there is a legal requirement to screen the certificate in restaurants and other public places.

New data. We collected the data from the certificates as follows. During the last wave of data collection (in November), we asked respondents reporting that they are vaccinated whether they had the certificate with them. If yes, we asked them to copy or type into our survey the text written in two specific text fields on their certificate (called Vaccine/Prophylaxis and Vaccine medical product in the Czech official app). For example, the text in these parts of the certificate says: SARS-CoV-2 mRNA vaccine, Comirnaty, Spikevax, Vaxzevria.

Results. 96.5% of those who reported being vaccinated confirmed that they had the certificate with them, and this proportion is very similar across conditions (Supplementary Table 14). This suggests that treated individuals are not more likely to avoid providing verifiable information. In contrast, if treated individuals were more prone to misreport, we would expect them to be more likely to report not having the certificate with them. Further, assessment of the typed-in text by independent raters suggests that, conditional on having the certificate, at least 94% of respondents actually used the certificate when responding to our detailed questions. Importantly, this rate is again very similar across conditions, suggesting that treated individuals were not more likely to misreport their vaccination status. In SI, we also show that respondents in the CONSENSUS condition were not more likely to make the effort and to find the required pieces of information on the Internet, by comparing their responses to information found in examples of certificates returned by google search.

Finally, we show that the estimated effect of the CONSENSUS condition on vaccine take-up is fully driven by greater prevalence of those whose possession of the certificate we are able to verify via questions about the certificate (Supplementary Table 15). We also show that, when we combine both verifications and consider vaccination status confirmed for those whose status we can verify through either the third-party verification survey or through the certificate verification, we arrive at the same conclusions.

Together, these new data and results give us confidence that the treatment effect does not arise due to misreporting. We also note that the observed dynamics of the treatment effect are hard to square with misreporting induced by experimenter demand. Experimenter demand effects are typically thought to potentially affect responses shortly after a treatment, whereas, in contrast, we find that the treatment effect on vaccination status emerges only gradually over a three-month period, as more people were becoming eligible to get vaccinated, and then it remained very stable.

Changes in the manuscript.

- We added a new sub-section “Verification of vaccination status” in the manuscript. In this sub-section we describe the new data collections and the results.
- We added a new Section 3.4. “Verification of vaccination status: methods and results” in Supplementary Information. This provides more details about the data collections, methods, and results.
- We added new Supplementary Tables 14 and 15 (see below).

Supplementary Table 14. Third party and certificate verification.

	(1)	(2)	(3)	(4)	(5)	(6)	(7)	(8)	(9)	(10)
	Observations	Response rate relative to Wave 11 sample			Verification rate for self-reported vaccinated			Verification rate for self-reported unvaccinated		
		CONSE	CONT	chi-sq	CONSE	CONT	chi-sq	CONSE	CONT	chi-sq
		NSUS	ROL	(p-value)	NSUS	ROL	(p-value)	NSUS	ROL	(p-value)
Panel A: Third party verification	1672	0.938	0.923	1.52 (0.218)	0.999	0.998	0.002 (0.967)	0.964	0.935	1.52 (0.218)
Panel B: Certificate verification	1364	0.960	0.970	0.99 (0.318)	0.941	0.949	0.473 (0.492)			

Notes: Column 1 reports observations for those who participated in wave 11 and [Panel A: participated in the third party verification, Panel B: reported being vaccinated with at least one dose of a vaccine against Covid-19]. Sample means in columns 2, 3, 5, 6, 8, 9. Columns 4, 7, and 10 report Pearson's chi-squared test F-statistic and a corresponding p-value in parentheses. Supplementary Information Section 3.4 describes both verification methods.

Supplementary Table 15. Effects of the CONSENSUS condition on take-up: More detailed analysis, based on whether the vaccination status verified (ordered and multinomial logit).

Specification	(1)	(2)	(3)	(4)	(5)	(6)
	Ordered logit			Multinomial logit		
Verification	Third party verification	Certificate verification	Third party OR certificate verification	Third party verification	Certificate verification	Third party OR certificate verification
Dependent variable	Vaccinated					
Waves 6-11, Effects of CONSENSUS on the prevalence of the following categories						
Vaccinated, verified	0.048*** (0.017) [0.004]	0.034** (0.017) [0.045]	0.038** (0.016) [0.016]	0.038** (0.016) [0.019]	0.030* (0.018) [0.093]	0.038** (0.016) [0.019]
Vaccinated, not verified	-0.005*** (0.002) [0.006]	-0.005** (0.002) [0.047]	-0.002** (0.001) [0.021]	-0.000 (0.007) [0.993]	0.008 (0.012) [0.520]	-0.000 (0.007) [0.993]
Not vaccinated	-0.043*** (0.015) [0.004]	-0.030** (0.015) [0.045]	-0.036** (0.015) [0.016]	-0.038** (0.015) [0.014]	-0.038** (0.015) [0.014]	-0.038** (0.015) [0.014]

Notes: Marginal effects for ordered logit (Columns 1-3) and multinomial logit (Columns 4-6) estimates. Delta-method standard errors in parentheses. P-values in square brackets. The dependent variable in all columns is a variable for vaccination take-up. The variable equals to 2 if the respondent reported having obtained at least one dose of a vaccine against Covid-19 and the self-report has been verified with either of the verification methods (See Supplementary Information Section 3.4 for more details on verification). It equals to 1 if the respondent reported having obtained at least one dose of a vaccine against Covid-19 but this has not been verified. It equals to 0 if the respondent reported not having obtained any vaccine against Covid-19. Full sample used. In all columns we use the pre-registered set of controls. All columns include wave fixed effects. Standard errors are clustered at an individual level.

2. One spec should be LASSO chosen controls – since they had to deviate from the pre-analysis plan.

RESPONSE

Thank you for the suggestion. In the revised version of the manuscript we now use two main specifications when reporting the results: one that controls for variables selected by the LASSO procedure, as you suggested, and one that controls for pre-registered variables. We document that the results do not rest on any specific choice of control variables, including these two and host of other specifications. The main steps and results are the following:

- We estimate the treatment effect when we control for variables selected by the LASSO procedure. This allows us to deal with the potential role of slightly imbalanced and non-pre-registered variables, and at the same time to tie our hands. In Figure 4 (see below), we show the effect for all waves separately, and find that the estimated treatment effect is around 4.5 p.p. for all waves implemented since the vaccine became available for all demographic groups (since July 2021). The estimated coefficients are also highly significant statistically (p-values range between 0.01 and 0.02).
- In Figure 5 (see below), we report results from pooled regressions to utilize data from all six waves implemented in July-November. We control for wave fixed effects and cluster standard errors at the individual level. The estimated treatment effect is 4.4 p.p. and it is highly statistically significant (p-value = 0.005).
- As the second main specification, we report results when we control for the set of pre-registered variables, following the suggestion of Referee 2. We also find a positive coefficient (3.5 p.p.), which is statistically significant at conventional levels (p-value = 0.026).
- Thus, the estimated coefficient is a bit lower for the pre-registered set of controls (by around 0.9 p.p.) as compared to the specification with LASSO-selected controls. This relatively small difference in the estimated effect size is to be expected because, as noted above, there is a small imbalance in (not pre-registered) variables that are highly predictive of the outcome (prior vaccination status and beliefs about trust of doctors) and that were selected to be appropriate control variables by the LASSO procedure. Figure 5 shows that adding these variables as controls increases the estimated effect size from 3.5 to 4.4 p.p.

In addition, we show that (i) the estimated coefficients are virtually unchanged when we control for different sets of observable characteristics (Figure 5), (ii) the results remain statistically significant at conventional levels when we calculate p-values using the randomization inference method (Supplementary Tables 9 and 12) and (iii) we arrive at very similar effect sizes as the LASSO specification when we use Difference-in-Difference estimator (Table 11).

Finally, the point estimates of the effect of the CONSENSUS condition, using the specifications listed above, are a bit larger for the fixed sample, i.e. those respondents who took part in all waves (right hand side of Figure 5 and Panel B of Supplementary Table 10). In the paper, we take a conservative approach to describing the results and focus mainly on the results for the full sample.

Together, this analysis shows that the CONSENSUS treatment has a robust positive impact on vaccination rates of 3.5 - 4.4 percentage points. Readers who believe that researchers should control for random imbalances in important baseline variables may favor the upper bound, while readers concerned about departures from pre-registered analyses may favor the lower bound.

Changes in the manuscript.

- We rewrote the sub-section “Effects on vaccination take-up” (p. 7-9) and pay much more attention to documenting and discussing the robustness of our findings, in terms of selection of control variables and attrition. In addition, our inclusion of data from the new waves collected during the fall allow us to perform additional analyses, and to draw stronger conclusions about the persistence of the effects.
- The estimates with controls selected by LASSO are reported in Figures 4 and 5, in Supplementary Figure 3 and Supplementary Tables 9, 10, and 12.

Figure 4. Effects of the CONSENSUS condition on vaccination take-up (Main Experiment). This figure plots the estimated effects of CONSENSUS by survey wave on getting at least one dose of a vaccine against Covid-19. We report the same four specifications as in Figure 3 (linear probability model with pre-registered controls using full (diamond) and fixed (triangle) samples, and double-selection LASSO linear regression selecting from controls in Supplementary Table 3 using full (square) and fixed (circle) samples). The whiskers denote the 95%-confidence interval based on Huber-White robust standard errors. In the upper part of the Figure, we report the timing, the total number of observations, and CONTROL mean for each wave. Supplementary Table 9 shows the regression results in detail.

Figure 5. Effects of the CONSENSUS condition on vaccine take-up: Robustness (Main Experiment). This specification chart plots the estimated effects of CONSENSUS on the likelihood of vaccine take-up for a pooled sample across Waves 6 to 11 (when the vaccine was available for all adults). All specifications include wave fixed effects. The darker (lighter) whiskers denote the 95% (90%)-confidence interval based on standard errors clustered at the respondent level. We report a range of specifications by sequentially adding sets of control variables in Supplementary Table 3. The main specifications are marked by blue diamonds. We report all specifications for both the full (left-hand side) and the fixed samples (right-hand side). Supplementary Table 10 shows the regression results in detail.

3. It was not clear how representative these doctors were. Supp table 1 is not exhaustive of population-level doctor characteristics and doesn't include p-values – there's also formatting issues. This could imply there is an element of deception in the "CONSENSUS" treatment if it's only picking up one part of the doctor distribution. Then in fact, the citizens are actually correct about the "true" trust in the vaccination across all Czech doctors. The need to address this.

RESPONSE

Thank you for pushing us to work harder on this. We have made two main steps.

First, we have managed to obtain more detailed summary statistics of the population of Czech doctors based on administrative data from the Institute of Health Information and Statistics (UZIS). This allows us to compare our sample of doctors with the whole population of doctors in the country based on virtually all observable characteristics of doctors that we collected in the survey (gender, 6 age categories, 3 levels of seniority, 6 categories of town sizes, 14 different regions).

- In Supplementary Table 1, we add a new column with statistics from UZIS (column 4) and columns that report tests of the equality of observable characteristics, both when we compare our sample with data provided by the Czech Medical Chamber (column 5) and by UZIS (column 6). We find that our sample has generally quite similar observable characteristics as the whole population of doctors. Although a number of differences tend to be statistically significant, this is partly due to the large sample size (N=9,650). Only for three variables that we compare (out of 30 comparisons with numbers from UZIS and 21 comparisons with numbers from the Czech Medical Chamber) is the difference larger than ten percentage points. Specifically, doctors who responded to our survey are somewhat more experienced (62% of doctors in our sample have more than 20 years of experience, while this number is 52% for the whole population), are more likely to work in towns with population >100k, and are less likely to work in towns with 20-100k.
- For each observable characteristic, we re-weight the responses in our sample using weights based on the UZIS data for the whole population of doctors. We report the opinions of doctors after this reweighting in the Supplementary Table 1. Given that we generally find very little variation in doctors' opinions across demographic characteristics, we arrive at very similar average opinions after the reweighting.

Second, according to official statistics from December 2021, 88% of doctors were actually vaccinated. We believe this number demonstrates the high demand of doctors for vaccination relatively well, since vaccination of doctors is not compulsory in the Czech Republic. This number confirms that a vast majority of doctors do indeed support vaccines, in line with the information we provide in the CONSENSUS condition.

Changes in the manuscript:

- We added data about how our sample compares to the population characteristics of doctors to Supplementary Table 1.
- We added text in the second paragraph in the section "Consensus of the medical community" (p. 4), where we say that the opinions of doctors in our survey are similar when we reweight responses based on true population shares, and that 90% percent support for vaccination is indeed in line with actual vaccine take-up among doctors in the country.

4. The magnitudes appear quite small in percent terms. The discussion should talk more about these magnitudes.

RESPONSE

Thank you (and Referee 2) for motivating us to talk more about the magnitudes. In our view, the point estimates of around 4 p.p. imply meaningful effect sizes, especially in light of the low costs of the intervention. Since 25-30% of respondents in the CONTROL condition were not vaccinated during the July-November period, the CONSENSUS condition reduces the number of those not vaccinated by around 13-16%.

In general, the most successful, low-cost behavioral nudges with documented impact on vaccine take-up report estimated effect sizes up to 5 p.p.^{2,3}, which is similar to the magnitude of the effect of providing information about consensus in doctors' opinions studied here. A noteworthy aspect of our study, besides its focus on a different type of intervention, is the documented persistence of the effects, which is another crucial margin for assessing the intervention effectiveness.

Further, we compare the magnitude of the observed effect with interventions that use information provision to signal the views of others, by providing truthful information about other people's vaccination intentions. Such type of intervention has been shown to increase intentions to get vaccinated by 1.9 p.p.⁴. Nudging health workers (not a representative sample) to get vaccinated by referring to vaccinated colleagues increases the likelihood of registering for vaccination by around 3 p.p.⁵. This suggests that provision of information from experts, compared to information from the general public, is likely to be particularly influential. But of course, this is only a across-study comparison, with all its limitations.

Changes in the manuscript

- We added a new paragraph in the Results section starting “The point estimates”, which discusses the estimated magnitude in the context of other, related studies (p. 8).

5. The survey doesn't include the script shown to the actual citizens (verbatim) about the CONSENSUS treatment.

RESPONSE

In the revised version, we have added the Czech version of the CONSENSUS treatment, in addition to the English translation of the treatment. Please see Section 3.3 in the Supplementary Information.

References

1. Haaland, I., Roth, C. & Wohlfart, J. Designing Information Provision Experiments. *J. Econ. Lit.* (2021). doi:10.2139/ssrn.3638879
2. Dai, H. *et al.* Behavioral Nudges Increase COVID-19 Vaccinations. *Nature* **597**, 404–409 (2021).
3. Milkman, K. L. *et al.* A megastudy of text-based nudges encouraging patients to get vaccinated at an upcoming doctor's appointment. *Proc. Natl. Acad. Sci. U. S. A.* **118**, 10–12 (2021).
4. Moehring, A. *et al.* Surfacing Norms to Increase Vaccine Acceptance. *SSRN Electron. J.* (2021). doi:10.2139/ssrn.3782082
5. Santos, H. C., Goren, A., Chabris, C. F. & Meyer, M. N. Effect of Targeted Behavioral Science Messages on COVID-19 Vaccination Registration among Employees of a Large Health System: A Randomized Trial. *JAMA Netw. Open* **4**, 11–14 (2021).

Referee 2

This paper examines a mismatch between doctors' beliefs about COVID-19 vaccines and the public's perceptions of those beliefs. This mismatch is then addressed by providing people with accurate information about doctors' beliefs. According to the analyses and discussion in this paper, this increases intentions to get vaccinated and actual subsequent (self-reported) vaccine take-up, even weeks later.

I quite liked the approach in this paper, including the collection of data about doctors' beliefs and then applying it in this survey. I hope that this has already encouraged propagating this information about doctors' beliefs more widely, at least in the Czech Republic, where this study was conducted.

RESPONSE

Thank you for your kind words. Our findings indeed raised the interest of the largest health insurance company in the country, which has recently begun to use the information about doctors' beliefs in some of its campaigns.

The point estimates of the effects are quite large. As a comparison point, I thought of work that instead invokes other "everyday people" as the reference group (Moehring et al. 2021, Santos et al. 2021). The former has a similar design, but finds significant though much smaller effects. The latter has a different setup (in that it uses text message reminder and short-term appointment signup as the outcome) and finds perhaps similarly sized effects. Anyway, the authors might think about highlighting that it seems like their "expert consensus" intervention is more effective than other "social norms" interventions. As someone who has worked on the latter, that is one thing that struck me in reading this. (Though also perhaps the present work provides only limited evidence about effect sizes, given the substantial uncertainty in the estimates.)

RESPONSE

Thank you for motivating us to compare the estimated effect sizes with the interesting experiments you refer to, and to some other related work. In the revised paper, we added the following paragraph in the Results section (p. 8): "The point estimates of around 4 p.p. imply a relatively large effect size, especially in light of the low costs of the intervention. Since the vaccination rate in the CONTROL condition was 70-75% during the July-November period, the CONSENSUS condition reduces the number of those not vaccinated by 13-16%. To compare, providing truthful information about other people's vaccination intentions was shown to increase intentions to get vaccinated by 1.9 p.p.²⁷. Nudging health workers to get vaccinated by referring to vaccinated colleagues increases likelihood of their registering for vaccination by around 3 p.p.²⁸. More generally, the most successful, low-cost behavioral nudges with documented impact on take-up have estimated effect sizes up to 5 p.p.^{6,7}, which is quite similar to the effect of providing information about consensus in doctors' opinions studied here. In addition, a noteworthy aspect of our study is the documented persistence of the effects, which is another crucial margin for assessing the intervention effectiveness."

Overall, I generally had a quite positive initial impression of this paper. And I think the whole thing makes intuitive and theoretical sense.

But I did have some concerns as I started digging into the details. These led me to wonder about the strength of the statistical evidence here. I elaborate on this below, but it seems important to better explore the robustness of the results to which covariates are adjusted for (particularly including the pre-registered analysis) and to probe questions about attrition.

Strength of evidence, imbalance, and adjustment for covariates

Contrary to the pre-analysis plan, the main analyses include adjustment for some additional covariates: “a non-pre-specified variable for being vaccinated in Wave0 and Wave0 beliefs about the views of doctors. We added the non-specified variables due to a detected imbalance in randomization.” (SI p. 32) These indeed seem like relevant covariates to adjust for. However, this kind of data-contingent adjustment is potentially worrying. If there were indeed a problem with randomization, one would want to get to the bottom of that. But I don’t see much evidence than anything was wrong; it is simply the case that there is a marginally significant imbalance ($.05 < p < .1$) in two covariates and a non-significant ($p > .1$) imbalance in another — without any correction for multiple hypothesis testing. This kind of data-contingent adjustment can increase error rates (e.g., Mutz et al. 2019), especially if no particular rule is followed, creating a “garden of forking paths” (Gelman & Loken 2014). Thus, unless the authors actually think randomization did not occur as planned (in which case perhaps more investigation is needed), I don’t see why these variables should be adjusted for in all main analyses. (Note also that there is no single obvious way to adjust for these covariates. The beliefs about doctors are often discussed in a dichotomous way, e.g., “Underestimating” vs “Overestimating” trust so one could imagine the adjustment being for that dichotomized version additionally or instead. This helps to create many possible specifications, and only one is reported.)

RESPONSE

Thank you very much for this comment and for clear suggestions. In the original version, our logic was that we need to control for variables that are highly predictive of the outcome, such as prior vaccination status or beliefs, and which, at the same time, happen not to be perfectly balanced, even if such baseline control variables are not pre-registered, since this may lead to biased estimates. But we understand your concern about this approach, since it may give researchers too much flexibility, and we show that our results do not rely upon this adjustment. In the revised version of the manuscript, we report two main specifications, in both of which we completely tie our hands: a pre-registered set of control variables, and control variables selected by LASSO. In addition, we report a host of robustness tests.

Here is how we proceed:

First, we have double-checked the procedures and the randomization was performed in a standard way (at the individual level by a computer). We have also performed an omnibus randomization test of joint significance for all covariates, as you suggested, and we cannot reject that the covariates do not differ across the CONSENSUS and CONTROL conditions (the p-value of this test is 0.342, see Supplementary Table 3).

Second, we show that finding a substantial positive impact of the CONSENSUS condition on vaccine take-up does not rest on a specific choice of control variables.

- As the first main specification, we estimate the treatment effect when using controls selected by the LASSO procedure, as suggested by Referee 1. This allows us to deal with the potential role of slightly imbalanced and non-preregistered variables, but keeps our hands tied. In Figure 4 (see below), we show the effect on take up for all waves separately, and we find that the point-estimates are around 4.5. p.p. for all the waves that were implemented after the vaccine became available for all demographic groups (since July 2021). The estimated coefficients are also highly significant statistically (p-values range between 1-2% levels). In Figure 5 (see below), we report the estimates of the effect with pooled data across all six waves implemented in July-November. We control for wave fixed

effects and cluster standard errors at the individual level. The estimated treatment effect is 4.4 p.p., and it is highly statistically significant (p -value = 0.005)

- As the second main specification, we estimate the effect when we control for the pre-registered set of control variables, following your suggestion in your comment below. In this specification, the measures of beliefs about doctors' opinions and own initial vaccination status are not controlled for. We also find a positive coefficient (around 3.5 p.p.), which is statistically significant at the 3% level in four out of six waves, and at the 10% level in two waves. Further, when we estimate the effect with pooled data across waves, the estimated effect size is 3.5 p.p. (p -value = 0.026).
- Thus, the estimated coefficient is around 0.8 of a percentage point lower for the specification with the pre-registered set of control variables than the specification with control variables selected by the LASSO procedure. This relatively small difference in the estimated effect size is to be expected, because there is a small imbalance in (not preregistered) variables that are highly predictive of the outcome (prior vaccination status and beliefs about doctor's trust in the vaccines) and that were selected as appropriate control variables by the LASSO procedure. Figure 5 shows that adding these variables as control variables increases the estimated effect size from 3.5 to 4.4 p.p.
- In addition, we show that (i) the estimated coefficients are virtually unchanged when we control for different sets of observable characteristics (Figure 5), (ii) the results remain statistically significant at conventional levels when we calculate p -values using the randomization inference method (Supplementary Tables 9 and 12) and (iii) we arrive at very similar effect sizes as with the LASSO specification when we use a Difference-in-Difference estimator (Supplementary Table 11).
- Next, the point estimates of the effect of the CONSENSUS condition, using the specifications listed above, are a bit larger for the fixed sample, i.e. those respondents who took part in all waves (right hand side of Figure 5 and Panel B of Supplementary Table 10). In the paper, we take a conservative approach to describing the results and focus mainly on the results for the full sample.
- For completeness, we tested the extent to which our estimates are sensitive to different ways of coding participant's beliefs about the views of doctors. We control for (i) continuous measures of beliefs (as in the original version of the manuscript) and (ii) a dummy variable indicating whether a respondent underestimated or overestimated doctors' views (as you suggested). Because the variation in the latter measure is very small as most of the respondents (90%) underestimated doctors' views, controlling for it instead of the continuous measure leads to slight reduction in the estimated coefficient (please see Columns 1 and 2 in Table R1 below), similarly as when we do not control for beliefs about doctors' views. For this reason, we also report the estimated when we control for (iii) a set of dummy variables indicating whether the respondent overestimated, heavily underestimated (above median) or slightly underestimated (below median) doctors' views. The estimated effect size is nearly identical to when we use the continuous measure (Columns 1 and 3). For the sake of space, we did not include this table in the revised manuscript, but we can add it, if recommended.

Together, this analysis shows that the CONSENSUS treatment has robust positive impacts on vaccination rates of 3.5 - 4.4 percentage points. Readers who believe that random imbalances in important baseline variables should generally be controlled for may prefer the point estimates based on specifications that control for these variables, such as the LASSO procedure. Readers with strong concerns about analysts' departures from pre-registered analyses may prefer to disregard the imbalance and focus on the specification with a pre-registered set of controls.

Changes to the manuscript:

- We rewrote the sub-section “Effects on vaccination take-up” (p. 7-9) and pay much more attention to documenting and discussing the robustness of our findings, in terms of selection of covariates.
- Supplementary Tables 3 and 4 report the results of the omnibus randomization test of joint significance for all covariates, for both the full and the fixed samples.
- All figures and tables in the manuscript and SI contain estimates with the pre-registered set of control variables and with the LASSO-selected control variables.
- Figure 5 and Supplementary Table 10 show the robustness of point estimates to different choices of covariates.
- Supplementary Table 11 reports the results of the difference-in-difference estimation.
- Supplementary Tables 9 and 12 report p-values using the randomization inference method.

Figure 4. Effects of the CONSENSUS condition on vaccination take-up (Main Experiment). This figure plots the estimated effects of CONSENSUS by survey wave on getting at least one dose of a vaccine against Covid-19. We report the same four specifications as in Figure 3 (linear probability model with pre-registered controls using full (diamond) and fixed (triangle) samples, and double-selection LASSO linear regression selecting from controls in Supplementary Table 3 using full (square) and fixed (circle) samples). The whiskers denote the 95%-confidence interval based on Huber-White robust standard errors. In the upper part of the Figure, we report the timing, the total number of observations, and CONTROL mean for each wave. Supplementary Table 9 shows the regression results in detail.

Figure 5. Effects of the CONSENSUS condition on vaccine take-up: Robustness (Main Experiment). This specification chart plots the estimated effects of CONSENSUS on the likelihood of vaccine take-up for a pooled sample across Waves 6 to 11 (when the vaccine was available for all adults). All specifications include wave fixed effects. The darker (lighter) whiskers denote the 95% (90%)-confidence interval based on standard errors clustered at the respondent level. We report a range of specifications by sequentially adding sets of control variables in Supplementary Table 3. The main specifications are marked by blue diamonds. We report all specifications for both the full (left-hand side) and the fixed samples (right-hand side). Supplementary Table 10 shows the regression results in detail.

Table R1: Effects of the CONSENSUS condition on the vaccine take up: Robustness – coding of beliefs about doctors’ views

	(1)	(2)	(3)
Dependent variable		Vaccinated	
Waves		31-36	
Panel A: Full sample			
Double-selection LASSO linear regression			
CONSENSUS	0.044*** (0.015)	0.037** (0.016)	0.044*** (0.015)
Controls		As in manuscript but belief questions dummy: Underest. / Overest.	As in manuscript but belief questions dummies: Heavy underest. / Slightly underest. / Overest.
Observations	As in manuscript 10,282	10,282	10,282
CONTROL mean	0.740	0.740	0.740
Panel B: Fixed sample			
Double-selection LASSO linear regression			
CONSENSUS	0.061*** (0.018)	0.056*** (0.019)	0.062*** (0.019)
Controls		As in manuscript but belief questions dummy: Underest. / Overest.	As in manuscript but belief questions dummies: Heavy underest. / Slightly underest. / Overest.
Observations	As in manuscript 7,272	7,272	7,272
CONTROL mean	0.743	0.743	0.743

I also find it a bit strange that the main text says: “Supplementary Table 3 provides evidence that the range of individual characteristics, including gender, age, household income, employment status, education, region and size of municipality, as well as vaccination intentions measured three weeks before the intervention are balanced across the experimental condition. The exceptions are ‘own vaccination’ and ‘prior beliefs about opinions of the doctors’ (measured in Wave0). Prior to the intervention, compared to participants in the CONTROL condition, the individuals in the CONSENSUS condition were less likely to be vaccinated themselves, and expected a smaller percentage of doctors to trust the vaccine and to intend to get vaccinated.” (p. 5) Looking at Table 3, it is a bit hard to see by what definition vaccination and prior beliefs about doctors are imbalanced but we don’t have imbalance in education and region (which have tests with $p < 0.1$). And it is also kind of hard to see how there is evidence of imbalance in own vaccination; if I am reading Table S3 correctly, the associated p -value here is 0.113 (i.e., not even $p < 0.1$). Now I guess this wouldn’t matter for creating further adjustment sets, since these variables are already in the preregistered set, but it does seem that this summary in the main text is at least confusing. Perhaps the authors have in mind some correction for the multiple categorical levels of education and region? More generally, I would suggest reporting a joint test of all of these covariates being randomized; presumably this retains the null. (Furthermore, it is far from clear to be that there is any real evidence of imbalance in these additional covariates, particularly conditional on everything else being adjusted for already.)

RESPONSE

Thank you for pointing this out. When mentioning the imbalance of beliefs and vaccination status in the original version, we implicitly took into account their predictive power of future vaccination status and the fact that they were not included in the pre-registered set. But you are right that this may have been confusing.

In the revised version, we provide the results of the test of joint significance of all covariates, as described above (Supplementary Tables 3 and 4). Further, we rewrote the paragraph describing the balance across the CONSENSUS and CONTROL conditions. We now say that there are no *systematic* differences in the pre-registered control variables. Further, we more clearly explain why we mention the slight imbalance in beliefs and prior vaccination status – because of their high predictive power and the fact that they were not preregistered – and include corresponding p-values directly into the text. Finally, we describe our two main regression specifications: one with a pre-registered set of control variables (guided by your comment), and one with control variables selected by the LASSO procedure, which selected beliefs and prior vaccination status to be among the appropriate set of controls.

Changes in the manuscript

- We rewrote the second paragraph on p. 6.
- Supplementary Tables 3 and 4 report the results of the omnibus randomization test of joint significance for all covariates, for both the full and the fixed samples.

As far as I can tell, the only variation on this analysis offered is one without controls altogether (for which the estimates is not statistically significant at $p < 0.05$, whether for vaccine demand in wave 1, Table S9, or vaccine takeup in wave 7, Table S11). That is, we don't ever get the preregistered analysis. This seems like an important omission, which should be corrected. This absence also seems at odds with the claim in the main text that "Second, the effects on vaccination demand and vaccine take-up are robust to changes in the set of control variables." Unless I am misunderstanding, again this only is true in the sense that there are some analyses entirely without covariates (e.g., column 2 of Table S11).

RESPONSE

As we explain above, the specification with the pre-registered set of controls is now one of the two main specifications and it is reported in all figures and tables which present the results. Thank you again for motivating us to do this. We also show that documenting the positive effect of the CONSENSUS condition on vaccine take-up is robust to using different sets of covariates (Figure 5).

Overall, this made me substantially less confident in the main results than I was initially. Perhaps this can be addressed by adding these additional analysis; I can't know. [I would also mention that, given the relatively large rank of the covariates, finite-sample bias may not be so small and so the authors might consider using an adjustment method with lower bias, as in Lin (2013) and Negi & Wooldridge (2021).]

RESPONSE

Thank you for this point. We agree that the set of pre-selected controls is relatively large (49 plus the CONSENSUS indicator), even though the number of observations still leaves us many degrees of freedom (the lowest number of observations in wave 6 (during summer holidays)

was 1,620). The fixed sample has 1,212 observations, which should also give us a sufficient number of degrees of freedom.

In the revised version, we further present pooled analyses across 6 waves (Figure 5 and Supplementary Tables 10 and 12). These analyses thus increase the number of degrees of freedom more than six times. We document a robust positive effect of the CONSENSUS condition on vaccine take-up with this specification. Next, sequentially adding controls or not using any controls barely changes the results (Supplementary Table 10), indicating that number of controls does not play an important role.

Finally, motivated by your suggestion and using the approach of Negi and Wooldridge (2021), we increase the number of controls by interacting de-meaned baseline covariates with the indicator for CONSENSUS condition. The results are very similar (please see Table R2 below). For the sake of space, we did not include this table in the revised manuscript, but we can add it, if recommended.

We hope that you share our sentiment that the new analyses and results much more credibly document that the CONSENSUS condition leads to a substantial increase in vaccine take-up. Your suggestions were truly helpful.

Changes in the manuscript:

- Figure 5 and Supplementary Table 10 report a range of specifications documenting the stability of results when changing numbers of covariates used.

Table R2: Effect of CONSENSUS on vaccine take up: Robustness test – adding control variables.

Dependent variable	(1)	(2)	(3)	(4)	(5)	(6)	(7)
Wave	6-11	6	7	8	9	10	11
Panel A: Full sample, Pre-registered set of controls interacted with CONSENSUS							
CONSENSUS					0.034**	0.040**	0.037**
	(0.015)	(0.018)	(0.017)	(0.017)	(0.017)	(0.018)	(0.016)
Controls	Pre-registered, interacted with CONSENSUS						
Observations	10,282	1,620	1,771	1,745	1,734	1,611	1,801
CONTROL mean	0.740	0.700	0.727	0.736	0.747	0.754	0.771
R-squared	0.347	0.372	0.350	0.352	0.367	0.350	0.335
Panel B: Fixed sample, Pre-registered set of controls interacted with CONSENSUS							
CONSENSUS					0.044**	0.043**	0.053**
	(0.021)	(0.022)	(0.022)	(0.021)	(0.021)	(0.021)	(0.021)
Controls	Pre-registered, interacted with CONSENSUS						
Observations	7,272	1,212	1,212	1,212	1,212	1,212	1,212
CONTROL mean	0.743	0.714	0.731	0.741	0.741	0.754	0.776
R-squared	0.391	0.405	0.403	0.387	0.400	0.392	0.386

Attrition

There is some attrition from the survey: “The response rate, as compared to the 187 base sample from Wave0, was 92% in Wave1 (March), 92% in Wave2 (April), 90% in Wave3 (May), 89% in Wave4 (May), 85% in Wave5 (June), 77% in Wave6 (July) and 84% in

189 Wave7 (August).” The authors note that the overall rate does not differ between treatment arms.

I think this analysis could be made more thorough. In particular, different average rates of attrition are not the only means by which attrition can bias results; differential attrition of different types of respondents is sufficient. So I would suggest conducting tests of differential attrition by baseline covariates, say, by a regression as in Table A4, but with interactions with covariates. A likely less powerful test could look at balance within the non-attrited sample. (In either case, this should be done so as to avoid problems of multiple comparisons, such as via a joint test.)

More generally, the authors might consider conducting some kind of sensitivity analysis or bounding exercise here to highlight how much of an issue this would all need to be to substantially alter the results.

RESPONSE

Thank you for motivating us to make the analysis of the potential role of attrition much more thorough. We now perform all the tests that you suggest and the results are re-assuring.

First, we find that participation rates in the CONSENSUS and CONTROL conditions do not differ on average, and that there is also no evidence of differential attrition by baseline covariates. We ran the suggested regressions with interaction effects between the CONSENSUS condition and all covariates. In Supplementary Table 13, we report the results of the omnibus test of joint significance of these interaction terms. The test does not reach significance at conventional levels in any of the eleven waves analyzed separately or when we focus on participation in all waves (being in the fixed sample). These results suggest that different types of individuals were not participating in follow-up waves in the CONSENSUS and CONTROL conditions.

Second, new Supplementary Table 4 provides a balance test for the non-attrited, fixed sample. We find that the balance is similar as that of the full sample. The estimated effect of the CONSENSUS condition on vaccination take-up is similar or, if anything, slightly larger in the non-attrited sample than in the full sample (Figures 4 and 5).

Third, as a sensitivity test, we impute missing vaccination status for those who did not participate in some of the waves, to be able to run an analysis on the full sample of respondents. We assume either that (i) their vaccination status has not changed since the last wave for which their data is available or that (ii) their status is the same as that reported in the soonest subsequent wave for which the data is available. The effect is robust (Supplementary Table 12, columns 3 and 4).

Changes in the manuscript:

- We added a new paragraph in the Results section summarizing the additional tests of the potential role of attrition described above (par. 2, p. 8).
- We expanded the Supplementary Table 13 testing the effect of the CONSENSUS condition on participation in each wave, by adding a new panel with p-values from a test of the joint significance of interaction effects between treatment and observable characteristics.
- We added a new Supplementary Table 4 with a balance test for the non-attrited/fixed sample.
- Supplementary Table 12, Columns 3 and 4 provide the results of the sensitivity analyses.

Heterogeneous effects

There is some discussion of heterogeneity of the effects. While it may be reasonable to say that the effects are “concentrated” in some particular subgroups, I think some of the discussion here commits the fallacy of comparing a statistically significant estimate and a non-significant one and assuming that this comparison is itself statistically significant. To provide evidence of heterogeneity per se, one should test that the effects in these subgroups are different. This can be done in a combined model with a Wald test. Or if the estimates are derived from entirely separate sets of data, as I believe all these analyses are, it can be done simply using the estimates and standard errors.

I did this for vaccine take-up for the subgroups formed by prior intention for Wave 7 (Table S11, Panel B, columns 9 and 10). The associated z-statistic is $(.090 - .023) / \sqrt{.041^2 + .016^2} = 1.52$. That is, the estimated effects in these two subgroups are not distinguishable. Eyeballing the rest of these, in Table S11 (which is the primary table supporting the claim in the main text, which is about take-up), I don't think any are significant.

Tables S13 and S14 present analyses disaggregated by many different covariates. These are described as analyses of heterogeneous effects; however, no tests of heterogeneity per se are conducted. Rather these are just tests of a number of conditional average treatment effects within each subgroup. So my comments above apply. I would also suggest that perhaps these numerous subgroup analyses (including those chosen to be highlighted in the main text) should somehow either (a) be subject to adjustment for multiple comparisons, (b) subsumed into some more flexible procedure for finding heterogeneous effects (e.g., causal forests), and/or (c) flagged as post hoc (particularly the ones in the main text).

RESPONSE

Agreed, in the revised manuscript we made the following changes:

- In all tables that present estimates of the effect of the CONSENSUS condition for different sub-groups, we now provide a formal test of the equality of the coefficients (Supplementary Tables 7, 12, and 17).
- We use more careful language in the paragraph in which we discuss the finding that the effects are concentrated among those who underestimated doctors' trust and among those who did not initially intend to get vaccinated (the last paragraph on p. 8). Specifically, at the beginning of the paragraph, we refer to this analysis as exploratory by saying: “The Supplementary Information describes exploratory analyses of how the treatment effect differs across different sub-samples of respondents (Supplementary Tables 7 and 12)”. Further, we conclude the paragraph with the following caveat: “Nevertheless, the analysis of heterogeneous effects should be treated as tentative because the differences in coefficients are not always statically significant and we do not adjust for multiple hypotheses testing.”
- Finally, we restrict the description of the heterogeneous treatment effects to the paragraph mentioned above and do not feature it in the abstract or in the concluding section.

Other comments:

I might suggest the authors use a term other than “experimental manipulation” throughout, particularly given the broad readership for this kind of work, as it can give a wrong, nefarious-sounding impression.

RESPONSE

Thank you for pointing out this negative connotation of the term. In the revised version, we replaced it with “information intervention”.

Some places (e.g., Figure 4) round p-values to 0.00. I would suggest they should instead either be stated as inequalities, using more digits, or using scientific notation. The latter, in particular, allows readers to more readily assess the strength of the evidence.

RESPONSE

Agreed, we now use more (three) digits in figures and tables where we report p-values.

Referee 3

The manuscript presents the findings from an experiment that tested the impact of a behavioral intervention -- informing people about the percent of doctors that support/trust COVID-19 vaccination – on vaccination uptake in the Czech Republic.

The information “nudge” was used because, as the authors describe, residents of the Czech Republic seemed to have an incorrect view, underestimating the percent of doctors that support/trust COVID-19 vaccination.

Some of the great features of the study:

- *It is a field experiment, randomized control trial,*
- *studying a non-US/UK sample,*
- *measuring behavior (vaccination uptake), and*
- *“longitudinal” – measuring the effect of the intervention over a period of roughly 5 months.*

I think this paper is important as we do not have many experimental findings about interventions that measure impact on COVID-19 uptake. However, the paper also has a few limitations that give me pause and that at minimum need to be made clearer to the readers of this paper as they assess the learnings / take-aways from this study:

- 1. We only seem to have participants’ self-reported vaccination status.*

RESPONSE

Thank you for this point. Given that vaccination status in our data is indeed self-reported, we agree it is useful to address in much greater depth the concern that the main effect could potentially arise due to the experimenter demand motivating some participants in the CONSENSUS condition to misreport being vaccinated even when they were not. Two types of new data help us to address the issue.

The first approach is based on data from another, independent entity that asked the same individuals about their vaccination status in a different survey. This data collection is guided by a comment from Referee 1 and by a recent methodological review paper¹ on information-provision experiments, suggesting that independently collected data are an effective way to deal with experimenter-demand effects. We refer to this new data collection as third party verification.

New data. We take advantage of the fact that different survey agencies have access to the panel our respondents are sampled from (the Czech National Panel). While the main data collection was implemented by one agency (NMS), we partnered with another agency (STEM/MARK) to include a question on vaccination status in a survey implemented on its behalf among the same sample. Since the survey agency, graphical interface, and topic of the survey were different from our main data collection, we believe respondents considered the two surveys to be completely independent of each other, and thus the experimenter demand effect potentially associated with our main survey is unlikely to affect responses in the third party verification survey. Out of 1,801 participants who took part in the last wave of the main data collection at the end November, 1,672 also took part in the third party verification survey, implemented two weeks later. This allows us to compare reported vaccination status at the individual level in the third-party verification survey and in our main survey for a vast majority of the sample.

Findings. We find several clear and reassuring patterns. First, mismatches in reporting are very rare in general and are not higher in the CONSENSUS condition. Only two

respondents (one in CONSENSUS and one in CONTROL) reported being vaccinated in the main survey but reported the opposite in the third-party verification survey (Supplementary Table 14; see below). In addition, 18 respondents (1%) reported being vaccinated in the verification survey but not in our survey. Note that the latter type of inconsistency does not necessarily imply misreporting, because the verification survey took place two weeks after the main survey and the respondents could have been vaccinated in the meantime. In any case, this type of mismatch in reporting is also not higher in the treatment than in the control condition.

Second, we show that the treatment effect is *not* driven by participants whose reports we are unable to verify because they did not take part either in the last wave of the main data collection or in the third-party verification survey. Specifically, using ordered and multinomial logit, we show that the effect of the CONSENSUS condition on lower prevalence of participants reporting not being vaccinated is almost fully explained by greater prevalence of those reporting being vaccinated and having their vaccination status verified in the third-party verification survey (Supplementary Table 15; see below).

The second approach to verify vaccination status aims to link reported vaccination status with official proofs of vaccination. Because we are unable to match the survey data with administrative data (which we have found is not feasible, despite our efforts to pursue this direction), we decided to take advantage of the fact that all vaccinated people receive an EU Digital COVID certificate issued by the Czech Ministry of Health and asked the respondents from our sample to provide specific pieces of information from their certificates. Importantly, the certificate contains information about the applied vaccine, which is unlikely to be known by someone without a certificate. Also, respondents should usually have the document readily available, typically in a mobile app, because there is a legal requirement to screen the certificate in restaurants and other public places.

New data. We collected the data from the certificates as follows. During the last wave of data collection (in November), we asked respondents reporting to be vaccinated whether they had the certificate with them. If yes, we further asked them to copy or type to our survey the text written in two specific text fields in their certificate (called Vaccine/Prophylaxis and Vaccine medical product in the Czech official app). For example, the text in these parts of the certificate says: SARS-CoV-2 mRNA vaccine, Comirnaty, Spikevax, Vaxzevria.

Results. 96.5% of those who reported being vaccinated confirmed that they had the certificate with them, and this proportion is very similar across conditions (Supplementary Table 14). This suggests that treated individuals are not more likely to avoid providing verifiable information. In contrast, if treated individuals were more prone to misreport, we would expect them to be more likely to report not having the certificate with them. Further, assessment of the typed-in text by independent raters suggests that, conditional on having the certificate, at least 94% of respondents actually used the certificate when responding to our detailed questions. Importantly, this rate is again very similar across conditions, suggesting that treated individuals were not more likely to misreport their vaccination status. In SI, we also show that respondents in the CONSENSUS condition were not more likely to make the effort and to find the required pieces of information on the Internet, by comparing their responses to information found in examples of certificates returned by google search.

Finally, we show that the estimated effect of the CONSENSUS condition on vaccine take-up is fully driven by greater prevalence of those whose possession of the certificate we are able to verify via questions about the certificate (Supplementary Table 15). We also show that, when we combine both verifications and consider vaccination status confirmed for those whose status we can verify through either the third-party verification survey or through the certificate verification, we arrive at the same conclusions.

Together, these new data and results give us confidence that the treatment effect does not arise due to misreporting. We also note that the observed dynamics of the treatment effect are hard to square with misreporting induced by experimenter demand. Experimenter demand effects are typically thought to potentially affect responses shortly after a treatment, whereas, in contrast, we find that the treatment effect on vaccination status emerges only gradually over a three-month period, as more people were becoming eligible to get vaccinated, and then it remained very stable.

Changes in the manuscript:

- We added a new sub-section “Verification of vaccination status” in the manuscript. In this sub-section we describe the new data collections and the results.
- We added a new Section 3.4. “Verification of vaccination status: methods and results” in Supplementary Information. This provides more details about the data collections, methods, and results.
- We added new Supplementary Tables 14 and 15.

Supplementary Table 14. Third party and certificate verification.

	(1)	(2)	(3)	(4)	(5)	(6)	(7)	(8)	(9)	(10)
	Observations	Response rate relative to Wave11 sample			Verification rate for self-reported vaccinated			Verification rate for self-reported unvaccinated		
		CONSE	CONT	chi-sq	CONSE	CONT	chi-sq	CONSE	CONT	chi-sq
		NSUS	ROL	(p-value)	NSUS	ROL	(p-value)	NSUS	ROL	(p-value)
Panel A: Third party verification	1672	0.938	0.923	1.52 (0.218)	0.999	0.998	0.002 (0.967)	0.964	0.935	1.52 (0.218)
Panel B: Certificate verification	1364	0.960	0.970	0.99 (0.318)	0.941	0.949	0.473 (0.492)			

Notes: Column 1 reports observations for those who participated in wave 11 and [Panel A: participated in the third party verification, Panel B: reported being vaccinated with at least one dose of a vaccine against Covid-19]. Sample means in columns 2, 3, 5, 6, 8, 9. Columns 4, 7, and 10 report Pearson’s chi-squared test F-statistic and a corresponding p-value in parentheses. Supplementary Information Section 3.4 describes both verification methods.

Supplementary Table 15. Effects of the CONSENSUS condition on take-up: More detailed analysis, based on whether vaccination status verified (ordered and multinomial logit).

Specification	(1)	(2)	(3)	(4)	(5)	(6)
	Ordered logit			Multinomial logit		
Verification	Third party verification	Certificate verification	Third party OR certificate verification	Third party verification	Certificate verification	Third party OR certificate verification
Dependent variable	Vaccinated					
Waves 6-11, Effects of CONSENSUS on the prevalence of the following categories						
Vaccinated, verified	0.048*** (0.017) [0.004]	0.034** (0.017) [0.045]	0.038** (0.016) [0.016]	0.038** (0.016) [0.019]	0.030* (0.018) [0.093]	0.038** (0.016) [0.019]
Vaccinated, not verified	-0.005*** (0.002) [0.006]	-0.005** (0.002) [0.047]	-0.002** (0.001) [0.021]	-0.000 (0.007) [0.993]	0.008 (0.012) [0.520]	-0.000 (0.007) [0.993]
Not vaccinated	-0.043*** (0.015) [0.004]	-0.030** (0.015) [0.045]	-0.036** (0.015) [0.016]	-0.038** (0.015) [0.014]	-0.038** (0.015) [0.014]	-0.038** (0.015) [0.014]

Notes: Marginal effects for ordered logit (Columns 1-3) and multinomial logit (Columns 4-6) estimates. Delta-method standard errors in parentheses. P-values in square brackets. The dependent variable in all columns is a variable for vaccination take-up. The variable equals to 2 if the respondent reported having obtained at least one dose of a vaccine against Covid-19 and the self-report has been verified with either of the verification methods (See Supplementary Information Section 3.4 for more details on verification). It equals to 1 if the respondent reported having obtained at least one dose of a vaccine against Covid-19 but this has not been verified. It equals to 0 if the respondent reported not having obtained any vaccine against Covid-19. Full sample used. In all columns we use the pre-registered set of controls. All columns include wave fixed effects. Standard errors are clustered at an individual level.

2. There seems to be an error of randomization on key variables. Specifically, the control and treatment group differ at baseline, that is, prior to intervention, in terms of

- their beliefs about doctors' opinions (trust and intentions to get vaccinated were significantly lower in the treatment group) and*
- the participants' own vaccination rates (lower in treatment group).*

While the authors control for these imbalances, this aspect gives me pause because (a) vaccination rates is also the key dependent variable later on, and (b) (mis)beliefs about doctors' opinions is supposedly the focal issue that drives vaccination rates.

RESPONSE

Thank you very much for this comment. In the revised version, we perform a thorough additional analysis focusing on this issue. We show that our finding of a positive treatment effect does not rely on controlling for the variables that happen to be slightly imbalanced, or more generally on the choice of control variables.

In Figure 5 and Supplementary Table 10, we show that the treatment effect is robust and statistically significant for various specifications.

- As one of the main regression specifications, we control for the set of pre-registered baseline variables, which does not include beliefs about doctors' opinions and own vaccination status. When we estimate the effect with pooled data across six waves implemented during a time when the vaccine was fully available (July-November; please see more on this below), the estimated effect size is 3.5 p.p., and the coefficient is

statistically significant (p -value=0.026). We also show that the estimated effect is similar in all six waves when they are analyzed separately (Figure 4; see below).

- As the second main specification, we use control variables selected by the LASSO procedure, as suggested by Referee 1. This approach allows us to take into account the potential role of slightly imbalanced baseline variables and at the same time to keep our hands tied. The estimated treatment effect across the waves is 4.4 p.p., and it is highly statistically significant (p -value = 0.005). Again, when we estimate the effect in each of the waves separately, we find very similar effects (Figure 4).
- Further, we perform a host of other specifications in which we change the set of control variables (Figure 5; see below the response to your next comment), including a specification in which we do not control for any covariates. Since the coefficient is also positive (3.6 p.p.), the results document that the main effect does not originate simply due to the vaccination rate in the CONSENSUS condition catching up with the CONTROL condition (more on this below).
- The effect also remains statistically significant at conventional levels when we calculate p -values using the randomization inference method (Supplementary Tables 9 and 12).
- We arrive at very similar effect sizes as with the LASSO specification when we use the data on vaccination rates at the baseline and employ the Difference-in-Difference estimation (Supplementary Table 11).
- Finally, the point estimates of the effect of the CONSENSUS condition, using the specifications listed above, are a bit larger for the fixed sample, i.e. those respondents who took part in all waves (right hand side of Figure 5 and Panel B of Supplementary Table 10). In the paper, we take a conservative approach to describing the results and focus mainly on the results for the full sample.

Together, this analysis shows that the CONSENSUS treatment has a robust positive impact on the vaccination rates of 3.5 - 4.4 percentage points. Readers who believe that researchers should control for random imbalances in important baseline variables may favor the point estimates based on specifications that control for these variables, such as the LASSO procedure. Readers who may be concerned about analysts' departures from pre-registered analyses may favor the specifications without these controls, such as the one with only pre-registered set of controls. Consequently, we use both of these specifications (with LASSO selected controls and with pre-registered set of controls) as the main specifications, and report the estimates in all corresponding figures and tables.

Finally, we also provide new evidence suggesting that the imbalance in those variables was an outcome of randomness rather than reflecting systematic differences in participants' characteristics across conditions. For more details on this point, please see our response to your next comment.

Changes to the manuscript.

- We rewrote the sub-section "Effects on vaccination take-up" (p. 7-9), and now pay much more attention to documenting and discussing the robustness of our findings, in terms of selection of covariates.
- All figures and tables in the manuscript and SI contain estimates with (i) the pre-registered set of control variables that do not include prior vaccination status and beliefs and (ii) the set of variables selected by the LASSO procedure that includes these variables.

- Figure 5 and Supplementary Table 10 show the robustness of point estimates to different choices of covariates.
- Supplementary Table 11 reports the results of the difference-in-difference estimation.

Figure 4. Effects of the CONSENSUS condition on vaccination take-up (Main Experiment). This figure plots the estimated effects of CONSENSUS by survey wave on getting at least one dose of a vaccine against Covid-19. We report the same four specifications as in Figure 3 (linear probability model with pre-registered controls using full (diamond) and fixed (triangle) samples, and double-selection LASSO linear regression selecting from controls in Supplementary Table 3 using full (square) and fixed (circle) samples). The whiskers denote the 95%-confidence interval based on Huber-White robust standard errors. In the upper part of the Figure, we report the timing, the total number of observations, and CONTROL mean for each wave. Supplementary Table 9 shows the regression results in detail.

In addition, the authors only consider that “these baseline differences could potentially contribute to underestimation of the treatment effects.” But what they do not consider is the possibility that because of this a-priori difference – even without an intervention – the participants in the CONSENSUS condition were more likely to catch up with the remaining population as more people became eligible for vaccination, likely knew more people that got it, and more information about the vaccines became widely known.

Particularly, given that the authors also point out “It is important to bear in mind that not everyone was able to get vaccinated from the very beginning of the data collection period. Different demographic groups became eligible to register for the vaccine at different points in time.”, it makes me wonder, if there are other reasons (other than the information nudge) that could explain why the CONSENSUS condition participants ended up having higher vaccination rates than those in the CONTROL group.

RESPONSE

Thank you for motivating us to think harder about alternative explanations for why the observed dynamics in the vaccination rates are different in the CONSENSUS and CONTROL conditions, other than their being driven by a genuine effect of the information provision, given that the prior vaccination rate was around 1 percentage point lower in the CONSENSUS than in the CONTROL condition.

The first concern could be a “reversion to the mean” in the CONSENSUS condition due to natural processes, resulting in vaccinations catching up with the vaccination rate in the CONTROL condition. We address this in the following way. We show that the treatment effect is predominantly driven by respondents in the CONSENSUS condition becoming *more* likely to be vaccinated than respondents in the CONTROL condition. This is documented by the following results:

- When we do not control for prior vaccination rate and beliefs, we still find a positive treatment effect (around 3.5 p.p.), as described above. Please see Figure 5 below, where we highlight all the relevant estimates.
- Further, we arrive at a nearly identical effect size even when we do not control for *any* observable characteristics - please see the first coefficient reported in Figure 5.

Therefore, even in the extreme case when “reversion to the mean” would fully explain the closing of the initial gap in the vaccination rate between the CONSENSUS and CONTROL conditions, rather than a genuine effect of the information, it would still account only for a small part of the overall effect of the CONSENSUS condition on vaccine take-up (less than 1 p.p.).

Changes in the manuscript

- We expanded the discussion of the robustness of the positive effect of the CONSENSUS condition on vaccine take-up, including the specifications in which the initial imbalance is not adjusted for (Results, last paragraph on p. 7).
- We added Figure 5 and Supplementary Table 10, which show the positive effect even when we do not control for prior vaccination status and beliefs.

Figure 5. Effects of the CONSENSUS condition on vaccine take-up: Robustness (Main Experiment). This specification chart plots the estimated effects of CONSENSUS on the likelihood of vaccine take-up for a pooled sample across Waves 6 to 11 (when the vaccine was available for all adults). All specifications include wave fixed effects. The darker (lighter) whiskers denote the 95% (90%)-confidence interval based on standard errors clustered at the respondent level. We report a range of specifications by sequentially adding sets of control variables in Supplementary Table 3. The main specifications are marked by blue diamonds. We report all specifications for both the full (left-hand side) and the fixed samples (right-hand side). Supplementary Table 10 shows the regression results in detail.

The second concern is that the initial imbalance in vaccination rates was not completely random and may have been correlated with other factors that could potentially cause differential dynamics across conditions, such as differences in characteristics that affect the timing of becoming eligible to get the vaccine, or differences in the degree of social contacts with people who got the virus. To address this concern, we take advantage of the fact that the data collection includes a module used by epidemiologists, to which we have gained access. This allows us to test whether there were baseline differences in the factors mentioned above across the conditions. The results do not support the alternative explanations. Specifically, we proceed as follows:

First, we focus on eligibility criteria. During the vaccine roll-out (until July 2021), there were two main types of individual characteristics that determined eligibility (for more details, please see Section 3.1 of Supplementary Information). The first criterion was based on age. We define exactly the same age categories that determined eligibility for vaccination at different points in time, starting from the oldest group: 70+, 65-69, 60-64, 55-59, 50-54,...3034, 18-29. The second type of criteria was based on an occupation - people working in health care, education, social services, and critical infrastructure were given earlier access to vaccination. We define workers in health care as those who reported that their International standard classification of occupations (ISCO) category is "Health professionals" (ISCO 22), workers in education as "Teaching professionals" (ISCO 23), and workers in social care as "Personal care workers in health services". (ISCO 532). Table R1 below shows that there are

no systematic differences across conditions in terms of the prevalence of different age categories that determined eligibility, or in the prevalence of priority occupations. Out of 13 comparisons, none is statistically significant at 10%. Further, as expected, controlling for these variables has virtually no effect on the main results. Since these variables were not pre-registered (at the time of pre-registration, the age cut-offs for vaccine eligibility were not known, and we pre-registered different age categories), and to economize on space, at this point we are inclined not to include this table in the paper, but we can of course add it if recommended.

Further, if the effects of the CONSENSUS condition were driven by differences in eligibility, we should observe the effect to dissipate once everyone became eligible for a vaccine. The new data from additional waves implemented in the fall 2021 are helpful to test this. The results reveal an opposite pattern – the effect of the treatment gradually increases as eligibility criteria were becoming more relaxed, and it persists during the period when the vaccine was fully available.

Second, we focus on the social environment and Covid-related experiences. We study the following variables measured at baseline: (i) number of social contacts, i.e., the number of people the respondent reported meeting for at least 5 minutes during the week prior to the interview, (ii) the number of potentially risky social contacts, i.e., the number of people the respondent reported meeting for at least 15 minutes without a face mask, (iii) whether the respondent was in personal contact with someone infected with Covid-19 during the prior two weeks, and (iv) whether the respondent personally knew someone who died because of Covid-19. For this set of variables, we also do not find any significant differences across the conditions – please see Table R2 below -- and the main effect is robust to controlling for them.

Table R1: Supplementary balance tests – eligibility criteria

	(1)	(2)	(3)	(4)
	Full sample	CONTROL	CONSENSUS	P-value
Age category	0.165	0.169	0.161	0.551
age cat 18-29	0.149	0.142	0.155	0.661
age cat 30-34	0.091	0.094	0.088	0.868
age cat 35-39	0.092	0.091	0.092	0.995
age cat 40-44	0.085	0.087	0.084	0.309
age cat 45-49	0.102	0.095	0.109	0.820
age cat 50-54	0.079	0.079	0.079	0.934
age cat 55-59	0.081	0.080	0.082	0.600
age cat 60-64	0.064	0.067	0.062	0.385
age cat 65-69	0.092	0.096	0.089	0.604
age cat 70+				
Occupation	0.007	0.006	0.009	0.436
health professionals	0.020	0.022	0.017	0.432
teaching professionals	0.009	0.010	0.007	0.345
personal care workers				

Table R2: Supplementary balance tests – risky contacts

	(1)	(2)	(3)	(4)
	Full sample	CONTROL	CONSENSUS	P-value
Risky contacts				
People met in past week for 5+ minutes	13.282	13.271	13.293	0.831
People met in past week, no face mask & >15 minutes	3.249	3.296	3.203	0.685
Met a covid positive person in past two weeks	0.060	0.059	0.062	0.779
Personally knows covid death case	0.198	0.205	0.192	0.484

3. It is puzzling to me that we only see significant effects on the key dependent variable vaccination uptake in the last 2-3 surveys. That is, if I understand right, the intervention is delivered in wave0 once, and only after three to five months do we see an effect of that onetime intervention. The authors do not comment on that enough and try to explain why that would be the case.

RESPONSE

Thank you for pointing this out. We believe that we have made important progress documenting that the CONSENSUS condition affects vaccination take-up not only during the last waves studied in the original version of the paper (in July-August), but that the effect also persists also far beyond the original timeframe. To do so, we implemented four new additional waves of data collection, monitoring vaccination take-up through September to the end of November.

Specifically, we find that the estimated effect is remarkably stable for the additional four months (Figure 4). In addition, in the September and November waves, we asked about respondent's intentions to get a booster (3rd) dose. The estimated effects are very similar in magnitude to the effects on take-up of the initial dose (around 4 p.p.), suggesting the information intervention elevates vaccination demand even nine months after it was implemented (Supplementary Figure 3).

We believe documenting such persistence is important. Since the vaccination demand in the CONTROL condition does not catch up with that in the CONSENSUS condition over such a long period, the results suggest that the type of vaccine hesitancy reduced by the CONSENSUS condition is resilient to policies, campaigns, and serious life disruptions that participants experienced during the fall. This includes a severe Covid-19 wave that took place in November 2021 in the Czech Republic, which resulted in one of the highest national mortality rates in global comparisons (see Section 3.1 of Supplementary Information).

We are not aware of any low-cost, behavioral intervention on Covid-19 vaccine take up that would be designed to measure or would document such long-term impacts. This further leverages the advantage of setting up a longitudinal data collection infrastructure, which you mentioned when summarizing our paper, and we hope these results will help to draw scholarly attention to this crucial, yet under-studied margin for assessing intervention efficiency.

Finally, we better clarify the reasons why we observe the treatment effect arising only gradually over the initial three-month period. The period studied can be divided into two stages. During the first stage (until June 2021), access to vaccinations was gradually rolled out due to binding constraints in terms of the amount of vaccine purchased by the government and its ability to distribute and administer the vaccine. During this period, supply shortages were gradually relaxed (Figure S6) and the eligibility rules changed regularly, to make the vaccine available for more demographic groups over time. In the second stage (since July),

vaccination has been freely available for the whole adult population. Section 3.1 in the Supplementary Information provides more contextual details, including the timeline of when different waves were implemented and of the eligibility criteria (Supplementary Table 5). We hope these contextual clarifications will help to explain why the full treatment effect on take-up manifests only with substantial delays, and thus help to strengthen the plausibility of the observed dynamics of the one-time intervention.

Changes in the manuscript:

- At several places in the paper, we describe the longer time span covered by the follow-up waves and the persistence of the CONSENSUS condition on vaccine take-up: in the abstract (p.1), Introduction (p. 3, last paragraph), Methods (p.5, par.2), and, in particular, in the Results section (p. 8, paragraphs 2 and 3).
- Figure 4 documents the dynamics of the treatment effect across all individual waves. We graphically distinguish the period when vaccination eligibility was restricted from the period when vaccines were freely available to all adults.
- We made a number of changes in the last paragraph on p. 5, to more clearly explain the context and the fact that eligibility restrictions were in place and were being gradually relaxed during the first four months after the intervention, while none of these restrictions were in place since July, which helps to explain the observed dynamics of the treatment effects.

4. Even if the participants represent a representative sample of the Czech Republic, how likely is the informational nudge intervention to be successful if rolled out to the entire population? I am wondering about that because the experiment was run in a special setting: a sample of individuals who agreed to be part of a panel and be interviewed regularly. In addition, it is not clear to me, how could the intervention tested in this manuscript be rolled out to the general population such that we could assume it would receive the same amount of attention as the panel participants devoted to the delivered survey? For example, if one were to send this the informational nudge about doctor's beliefs via informational mail-in flyers from the government or media ads, could we just assume that that would be successful given the experiment described in this manuscript? I am doubtful.

Together, while I admire the authors' efforts and do think it is important that we gather more information on the effectiveness of easy-to-scale behavioral interventions for vaccination uptake, I am not sure how much we can learn about whether the observed effects described here are truly due to the one-time intervention and could we expect if scaled (and not clear how one could scale it).

RESPONSE

Thank you for this point. We totally agree that the level of attention is an important factor in determining the success of any campaign that is based on information provision, including those potentially building on insights from this RCT. Consequently, it is important to carefully choose the mode of delivery of the information when designing large-scale campaigns. We can imagine that well-designed campaigns could attract even more attention than “cold” provision of information as in our experiment, by, for example, providing repeated exposure to the information, or by employing more engaging graphics than are allowed in standard survey interfaces. At the same time, given (i) the low-cost nature of the treatment, (ii) its ability to tackle the type of hesitancy that is resilient to standard campaigns and life disruptions (at least those implemented and experienced in the Czech Republic) and (iii) large-scale implementation, the expected benefits are still very large even if real-life campaigns attract

somewhat less attention than in our survey. In principle, such campaigns should also be relatively easy to implement, by, for example, sending flyers by mail or e-mails, or presenting the information during TV or radio spots, or on billboards.

Moreover, given that this study not only shows that the intervention “works” by increasing take-up, but also sheds light on a specific mechanism driving these effects, the cost-efficiency can be boosted by targeting the right populations. For instance, given that the effect of the CONSENSUS condition is driven, both in theory and empirically, by changes in beliefs and take-up among those who initially did *not* want to get vaccinated, insurance companies or governmental agencies with access to information about vaccination status could increase the cost efficiency of an information campaign by targeting this sub-group.

In this context, it is noteworthy that the “nudge” studied in this paper can be seen as complementary to other types of low-cost “behavioral” interventions with documented impacts – great examples are the following two recent RCTs^{2,3}. The focus of most behavioral interventions is on helping people who already want to get vaccinated to follow through on their intentions (e.g., by simplifying registration, sending reminders, changing default actions, etc.). In contrast, our work illuminates a low-cost approach to increasing take-up among those who initially do *not* intend to get vaccinated. Consequently, an exciting question for future research is whether combining a CONSENSUS nudge, similar to the one studied here, with behavioral nudges like those studied in the papers mentioned above could further boost their impact, and thus broaden the applicability of behavioral science interventions.

Changes in the manuscript:

- We added a discussion about the importance of attracting a sufficient level of attention when this intervention is scaled up and used in real-life campaigns (Discussion section, last paragraph, p. 10).
- We would be very happy to provide more discussion related to scaling up this intervention, along the lines described in the last two paragraphs in our response, if space allows and we are advised to do so.

References

1. Haaland, I., Roth, C. & Wohlfart, J. Designing Information Provision Experiments. *J. Econ. Lit.* (2021). doi:10.2139/ssrn.3638879
2. Milkman, K. L. *et al.* A megastudy of text-based nudges encouraging patients to get vaccinated at an upcoming doctor’s appointment. *Proc. Natl. Acad. Sci. U. S. A.* **118**, 10–12 (2021).
3. Dai, H. *et al.* Behavioral Nudges Increase COVID-19 Vaccinations. *Nature* **597**, 404–409 (2021).

Reviewer Reports on the First Revision:

Referees' comments:

Referee #1 (Remarks to the Author):

The authors have done a superb job addressing my specific concerns.

Referee #2 (Remarks to the Author):

The revised paper addresses some issues raised by myself and other reviewers. I remain enthusiastic about this work. The duration of the survey -- expanded in this revision as well -- is helpful in documenting the relatively long-run gap in take-up caused by the intervention, compared with other valuable work in this area, e.g., Dai et al. (ref 7).

Covariates and randomization

I am glad to see the additional specifications including covariates, including the preregistered set and the lasso-based selection. (Though I don't really know that the double-selection lasso is needed or well-motivated here given that this is a randomized experiment; perhaps a better reference is Bloniarz et al. 2016.)

The writing about the covariates and posited imbalance still seems not quite right to me -- and likely to confuse some readers. For example:

"Nevertheless, we note that three potentially important but not pre-registered variables are not perfectly balanced. Since these three variables are highly predictive of vaccination take-up, not controlling for them could potentially bias the estimation of treatment effects, as is also indicated by the LASSO procedure, which selects these variables among a set of variables that should be controlled for in our estimates." (p. 6)

First, what does "not perfectly balanced" mean here? My guess is that all of the variables are not perfectly balanced, as perfect balance would be having identical numbers of subjects with each value in treatment and control, and would typically only be achieved in the blocked/stratified randomization.

Second, in what sense is does this "bias the estimation of treatment effects"? On typical theoretical analyses of randomized experiments, as long as we believe randomization occurred as planned, error due to random differences between groups is not bias; it is **variance** and is correctly accounted for in statistical inference.

This is also related to Reviewer 3's review. I think it is important for the authors to avoid the incorrect interpretation that something went wrong with their randomization. All indications are that it occurred exactly as planned. However, there can be substantial precision gains from adjusting for covariates, so this provides a reason to prefer the covariate-adjusted estimates.

If I was going to write this paragraph, I would say something like:

Nevertheless, because the randomization was not stratified (i.e. blocked) on baseline covariates, there are random imbalances in covariates, as expected. Some of the larger differences are variables that were not specified in the pre-registered set of covariates to use for regression adjustment: [stating the covariates, I might suggest reporting standardized differences, not p-values here].

Of course, the paper is the authors' to write, but I would just advise that unless they have a reason to believe the randomization did not occur as expected (not just that there were random differences in some covariates), they should avoid giving readers this impression.

Demand effects

Other reviewers seem more concerned about the potential for demand effects. I was less concerned about this given the multi-wave nature of the survey. And I think the authors responses further reduce my worries here.

Attrition

Differential attrition can be a worry in this setting. I think I like the approach the authors have taken for imputing the missing vaccination status among respondents who are missing some data. However, I wish a bit more detail was provided here. For example, what causes the different sample sizes here? Maybe I could work this out, but I think it is better for the authors to make this more explicit.

Representativeness

There are some questions about whether the sample here is representative of a larger population, including on unobserved traits that might affect receptivity to this informational intervention; this is raised by Reviewer 3. I don't know that there is any simple fix here, but perhaps it is relevant to look at and perhaps cite this recent paper (Bradley et al. 2021) on remaining bias in some related surveys, even after weighting.

Overall, I think this paper makes a valuable contribution and the revisions, particularly around adjustment for covariates, make me more confident in the results.

Sincerely,
Dean Eckles

Bloniarczyk, A., Liu, H., Zhang, C. H., Sekhon, J. S., & Yu, B. (2016). Lasso adjustments of treatment effect estimates in randomized experiments. *Proceedings of the National Academy of Sciences*, 113(27), 7383-7390.

Bradley, V. C., Kuriwaki, S., Isakov, M., Sejdinovic, D., Meng, X. L., & Flaxman, S. (2021).

Unrepresentative big surveys significantly overestimated US vaccine uptake. *Nature*, 600(7890), 695-700.

Referee #3 (Remarks to the Author):

I am impressed by the additional data collection efforts and the analyses the authors conducted to confirm the robustness and persistence of their intervention's effect. My concerns have been largely addressed. I commend the authors for their thorough revision.

I would recommend the authors add a paragraph on some potentially necessary requirements that would need to be in place in order to successfully scale the success of their intervention. For example, research in advertising and communication has shown that trust in the source of information is important. At minimum, I believe the authors would want to acknowledge that if one were to try and scale this intervention, there would need to exist a trusting source. In that sense the recent publications the authors cite (Milkman et al. 2021 as well as Dai et al. 2021) are different from the current draft in that Milkman et al. and Dai et al. tested SMS interventions sent out by patients' own health care systems -- presumably trusted sources (and sources that had the ability to reach out to individual patients as they had those patients' emails/phone numbers).

Finally, the authors may want to consider citing the Milkman et al. 2022 paper with Walmart customers, which represents a much larger and more diverse sample and thus, a higher powered study.

Author Rebuttals to First Revision:

Referee #1 (Remarks to the Author):

The authors have done a superb job addressing my specific concerns.

RESPONSE

We are glad to hear that we satisfactorily addressed your concerns. Again, thank you very much for extremely thoughtful, thorough, and constructive comments.

Referee #2 (Remarks to the Author):

The revised paper addresses some issues raised by myself and other reviewers. I remain enthusiastic about this work. The duration of the survey -- expanded in this revision as well -- is helpful in documenting the relatively long-run gap in take-up caused by the intervention, compared with other valuable work in this area, e.g., Dai et al. (ref 7).

Covariates and randomization

I am glad to see the additional specifications including covariates, including the preregistered set and the lasso-based selection. (Though I don't really know that the double-selection lasso is needed or well-motivated here given that this is a randomized experiment; perhaps a better reference is Bloniarz et al. 2016.)

The writing about the covariates and posited imbalance still seems not quite right to me -- and likely to confuse some readers. For example: "Nevertheless, we note that three potentially important but not pre-registered variables are not perfectly balanced. Since these three variables are highly predictive of vaccination take-up, not controlling for them could potentially bias the estimation of treatment effects, as is also indicated by the LASSO procedure, which selects these variables among a set of variables that should be controlled for in our estimates." (p. 6)

First, what does "not perfectly balanced" mean here? My guess is that all of the variables are not perfectly balanced, as perfect balance would be having identical numbers of subjects with each value in treatment and control, and would typically only be achieved in the blocked/stratified randomization.

Second, in what sense is does this "bias the estimation of treatment effects"? On typical theoretical analyses of randomized experiments, as long as we believe randomization occurred as planned, error due to random differences between groups is not bias; it is **variance** and is correctly accounted for in statistical inference.

This is also related to Reviewer 3's review. I think it is important for the authors to avoid the incorrect interpretation that something went wrong with their randomization. All indications are that it occurred exactly as planned. However, there can be substantial precision gains from adjusting for covariates, so this provides a reason to prefer the covariate-adjusted estimates.

If I was going to write this paragraph, I would say something like: Nevertheless, because the randomization was not stratified (i.e. blocked) on baseline covariates, there are random imbalances in covariates, as expected. Some of the larger differences are variables that were not specified in the pre-registered set of covariates to use for regression adjustment: [stating the covariates, I might suggest reporting standardized differences, not p-values here].

Of course, the paper is the authors' to write, but I would just advise that unless they have a reason to believe the randomization did not occur as expected (not just that there were random differences in some covariates), they should avoid giving readers this impression.

RESPONSE

Thank you very much for your detailed comments and specific recommendations how to further improve our description of randomization and the decision to use LASSO as one of the main specifications. In the revised version, we closely follow your suggestions. Specifically, on page 5 we say the following and we also cite the study by Bloniarz et al. 2016 (#29):

“Extended Data Table 1 and Supplementary Table 3 show no systematic differences in the set of baseline characteristics pre-registered as control variables. Nevertheless, because the randomization was not stratified on baseline covariates, there are random imbalances in some covariates, as expected. Some of the larger differences are for variables not included in the set of pre-registered control variables. Specifically, prior to the intervention, compared to participants in the CONTROL condition, the individuals in the CONSENSUS condition were slightly less likely to be vaccinated themselves (standardized mean difference (SMD)= 0.069), and expected a smaller percentage of doctors to trust the vaccine (SMD=0.072) or to intend to get vaccinated (SMD=0.090). Since these three variables are highly predictive of vaccination take-up, we report two main regression specifications: (i) with the preregistered set of control variables and (ii) with control variables selected by the LASSO procedure²⁹. To document robustness, we also report estimates with no control variables and with alternative sets of control variables.”

Demand effects

Other reviewers seem more concerned about the potential for demand effects. I was less concerned about this given the multi-wave nature of the survey. And I think the authors responses further reduce my worries here.

RESPONSE

Thank you, we are glad to hear that.

Attrition

Differential attrition can be a worry in this setting. I think I like the approach the authors have taken for imputing the missing vaccination status among respondents who are missing some data. However, I wish a bit more detail was provided here. For example, what causes the different sample sizes here? Maybe I could work this out, but I think it is better for the authors to make this more explicit.

RESPONSE

On page 7, we added the following text (in bold) when describing how we impute the missing vaccination status, which clarifies why the sample sizes differ:

*“As a sensitivity test, we impute missing vaccination status for those who did not participate in some of the waves and assume either that (i) their vaccination status has not changed since the last wave for which the data is available or that (ii) their status is the same as the one reported in the earliest next wave for which the data is available. **The first approach allows us to impute all the missing information because we know each participant’s vaccination status in the initial wave. The second approach allows us to impute the missing information, except in cases when a respondent did not participate in the last wave.**”*

Representativeness

There are some questions about whether the sample here is representative of a larger population, including on unobserved traits that might affect receptivity to this informational intervention; this is raised by Reviewer 3. I don't know that there is any simple fix here, but perhaps it is relevant to look at and perhaps cite this recent paper (Bradley et al. 2021) on remaining bias in some related surveys, even after weighting.

RESPONSE

Thank you for motivating us to read the study of Bradley et al. 2021, which shows that some (convenience) online surveys (e.g., via Facebook) do not provide a representative picture of attitudes to vaccination, even after re-weighting based on observables. In the revised version, we cite this paper and show that this is not the case in our study. When we take observations from the CONTROL condition, we find that the levels and dynamics of vaccination take up very closely mimic overall vaccination take up in the country among adults. This suggests that vaccination attitudes measured in our study are likely representative of the vaccination rates of a larger population. Although we do not see any direct empirical fix to address a subtler concern that our sample might not be representative in terms of unobservable characteristics affecting receptivity to the information treatment studied, we think the fact that we can document that overall vaccination attitudes are representative attenuates such concerns. On p. 4 we include the following text with a reference to Bradley et al. 2021 (#28):

“In addition, the vaccination rate reported in our sample closely mimics the levels and dynamics of the overall adult vaccination rate in the country (Extended Data Fig. 1). This comparison suggests that attitudes to vaccination in our sample are likely to be representative of the larger population, in contrast to surveys based on convenience samples.²⁸ Although this pattern is reassuring, we cannot

test and fully rule out a possibility that our sample might not be representative in terms of unobservable characteristics affecting receptivity to the information treatment studied.”

Overall, I think this paper makes a valuable contribution and the revisions, particularly around adjustment for covariates, make me more confident in the results.

RESPONSE

Thank you very much for extremely thoughtful, thorough, and constructive comments.

Sincerely,
Dean Eckles

Bloniarz, A., Liu, H., Zhang, C. H., Sekhon, J. S., & Yu, B. (2016). Lasso adjustments of treatment effect estimates in randomized experiments. *Proceedings of the National Academy of Sciences*, 113(27), 7383-7390.

Bradley, V. C., Kuriwaki, S., Isakov, M., Sejdinovic, D., Meng, X. L., & Flaxman, S. (2021). Unrepresentative big surveys significantly overestimated US vaccine uptake. *Nature*, 600(7890), 695-700.

Referee #3 (Remarks to the Author):

I am impressed by the additional data collection efforts and the analyses the authors conducted to confirm the robustness and persistence of their intervention's effect. My concerns have been largely addressed. I commend the authors for their thorough revision.

RESPONSE

We are glad to hear that we satisfactorily addressed your concerns. Again, thank you very much for extremely thoughtful, thorough, and constructive comments.

I would recommend the authors add a paragraph on some potentially necessary requirements that would need to be in place in order to successfully scale the success of their intervention. For example, research in advertising and communication has shown that trust in the source of information is important. At minimum, I believe the authors would want to acknowledge that if one were to try and scale this intervention, there would need to exist a trusting source. In that sense the recent publications the authors cite (Milkman et al. 2021 as well as Dai et al. 2021) are different from the current draft in that Milkman et al. and Dai et al. tested SMS interventions sent out by patients' own health care systems -- presumably trusted sources (and sources that had the ability to reach out to individual patients as they had those patients' emails/phone numbers).

RESPONSE

We agree, having a trusted source of information and access to a large number of contacts is likely very important. We expanded the concluding paragraph of the paper, in which we discuss several aspects that policy-makers and organizations aiming to scale up our intervention need to consider, including the ability to attract a sufficient degree of attention, providing the information by a trusted source and involving an organization with access to contacts for a large number of individuals. Natural candidates for organizing such information campaigns would include Health Ministries, health insurance companies, and health care providers, since they are often trusted in many settings and have patient contact information.

Specifically, on page 9 we expanded the text in the following paragraph:

“To guide efforts to scale up this intervention, we discuss what types of factors may affect its efficiency and how we view the boundary conditions in terms of the intervention’s applicability beyond the context we studied. We estimate the effects of a one-time intervention, among a sample in which most people likely paid attention to the information. Understanding whether the efficiency of the intervention can be fostered by repeated provision of information, as some research suggests³⁷, and which modes of delivery, such as media ads, text messages, or informational mail flyers can best attract a sufficient degree of attention is an important next step for future research. Next, in many settings, implementing such information campaigns by governments, health insurance companies, or health care providers may help to facilitate access to the contacts of large numbers of individuals^{4,5} and to address the need for a trusted source to provide the information intervention....”

Finally, the authors may want to consider citing the Milkman et al. 2022 paper with Walmart customers, which represents a much larger and more diverse sample and thus, a higher powered study.

RESPONSE

Thank you for suggesting that we cite the recent Milkman et al. 2022 paper on behavioral nudges aiming to increase take up of flu vaccinations. We enjoyed reading it. In the last paragraph of our paper (copied and pasted above), we added a reference to Milkman et al. 2022 (#37) when discussing the possibility that repeated exposure to information or behavioral nudges may increase their efficiency in causing a behavioral change.